# On the Out-of-Distribution Generalization of Self-Supervised Learning

**Wenwen Qiang** [1 2 *]   **Jingyao Wang** [1 2 *]   **Zeen Song** [1 2]   **Jiangmeng Li** [1 2]   **Changwen Zheng** [1 2]

## Abstract

In this paper, we focus on the out-of-distribution (OOD) generalization of self-supervised learning (SSL). By analyzing the mini-batch construction during the SSL training phase, we first give one plausible explanation for SSL having OOD generalization. Then, from the perspective of data generation and causal inference, we analyze and conclude that SSL learns spurious correlations during the training process, which leads to a reduction in OOD generalization. To address this issue, we propose a post-intervention distribution (PID) grounded in the Structural Causal Model. PID offers a scenario where the spurious variable and label variable is mutually independent. Besides, we demonstrate that if each mini-batch during SSL training satisfies PID, the resulting SSL model can achieve optimal worst-case OOD performance. This motivates us to develop a batch sampling strategy that enforces PID constraints through the learning of a latent variable model. Through theoretical analysis, we demonstrate the identifiability of the latent variable model and validate the effectiveness of the proposed sampling strategy. Experiments conducted on various downstream OOD tasks demonstrate the effectiveness of the proposed sampling strategy.

## 1. Introduction

Self-supervised learning (SSL) has emerged as a powerful paradigm for training models without relying on labeled data. Meanwhile, SSL models have achieved competitive or superior performance on various downstream tasks compared to supervised learning approaches (Chen et al., 2020; Grill et al., 2020a; Zbontar et al., 2021; He et al., 2022; Tong et al., 2022). However, despite their superior performance,

*Equal contribution [1]Institute of Software Chinese Academy of Sciences, Beijing, China [2]University of the Chinese Academy of Sciences, Beijing, China. Correspondence to: Jiangmeng Li <jiangmeng2019@iscas.ac.cn>.

*Proceedings of the 42nd International Conference on Machine Learning*, Vancouver, Canada. PMLR 267, 2025. Copyright 2025 by the author(s).

SSL models face significant challenges in generalizing to out-of-distribution (OOD) data. Understanding and improving the OOD generalization capabilities of SSL is crucial for deploying these models in real-world scenarios where the data distribution can shift over time.

To investigate the OOD generalization properties of SSL, we propose examining the batch construction process during SSL training. SSL methods are generally categorized into two main types: discrimination-based SSL (D-SSL) (Chen et al., 2020; Grill et al., 2020a) and generation-based SSL (G-SSL) (He et al., 2022; Tong et al., 2022). The core principle of D-SSL is augmentation invariance, ensuring that the feature representations of two different augmentations of the same sample are similar. In contrast, G-SSL focuses on the mask and reconstruction principle, where a portion of a sample is masked and then reconstructed using an encoder-decoder structure. Leveraging these principles, augmented samples derived from the same original sample, as well as samples before and after masking, can be considered anchor-related pairs. During SSL training, each pair is treated as a distinct class, effectively framing each mini-batch as a multi-class learning task. Consequently, the SSL training process can be perceived as learning a distribution over tasks based on discrete training tasks, enabling the trained SSL model to generalize to new, unseen tasks, thus demonstrating its OOD generalization capability. However, machine learning is prone to learning spurious correlations that vary between classes or environments (Wang et al., 2023a; 2022). Therefore, although SSL is highly effective in OOD generalization, it may still face the challenge of mitigating spurious correlations.

Building upon the analysis in Section 3, we examine the aforementioned challenge from the perspectives of data generation and causal inference. First, we conclude that the similarity or reconstruction between samples within a pair is affected by several unobservable factors, such as background or texture information independent of the foreground. We also find that the spurious correlation between the anchor and the unobservable variable can vary with the tasks, making it difficult to eliminate it using the unified causal criterion proposed by (Pearl et al.; Pearl, 2009). Furthermore, we demonstrate that, under these circumstances, the SSL model learns to measure similarity or reconstruct using spurious causal factors. This reliance leads to a lack of discriminabil-

ity within each mini-batch task, preventing the SSL model from effectively learning the true task distribution and consequently resulting in diminished OOD generalization. To address this issue, we define a new distribution called the post-intervention distribution (PID), characterized by mutual independence between the unobservable variable and the anchor. We demonstrate that when the task distribution adheres to PID, the SSL model trained under this condition achieves the lowest worst-case risk, thereby attaining optimal worst-case OOD performance. This insight motivates us to design a new mini-batch sampling strategy that ensures the resulting mini-batches satisfy PID constraints, thereby enhancing the OOD generalization capability of SSL.

Based on the above analysis and discussion, we propose a novel mini-batch sampling strategy consisting of two stages. In the first stage, we aim to learn a latent variable model to capture the correlations between different variables, i.e., conditional distributions. We prove the identifiability and uniqueness of the resulting latent variable model under a given equivalence relation. In the second stage, we propose a sufficient condition to obtain the balancing score. Using this, we obtain the mini-batch samples through balancing score matching. We also provide a theoretical guarantee that the mini-batches obtained by the proposed sampling strategy approximately satisfy the PID.

In summary, we make the following contributions: **1**) Analysis of SSL Batch Construction: We provide a detailed analysis of how mini-batch construction in SSL influences OOD generalization; **2**) Causal Framework for SSL: We introduce a causal framework to understand and mitigate the impact of spurious correlations on SSL models; **3**) PID-Based Sampling Strategy: We propose a theoretically grounded mini-batch sampling strategy that ensures the generated batches conform to PID, improving OOD performance; **4**) Empirical Validation: We validate our approach through extensive experiments, demonstrating significant improvements in OOD generalization across multiple tasks. The whole logical structure of this paper is further clarified in **Appendix** I

## 2. Revisiting SSL from a Pairwise Perspective

In the training phase, the training data is structured into mini-batches, with each mini-batch denoted as $X_{tr} = \{x_i\}_{i=1}^N$, where $x_i$ represents the $i$-th sample and $N$ is the total number of samples. In D-SSL methods such as SimCLR (Chen et al., 2020) and Barlow Twins (Zbontar et al., 2021), each sample in $X_{tr}$ undergoes stochastic data augmentation to generate two augmented views, e.g., for $x_i \in X_{tr}$, the augmented samples can be represented as $x_i^1$ and $x_i^2$. For G-SSL methods, like MAE (He et al., 2022) and VideoMAE (Tong et al., 2022), $x_i$ is first divided into multiple small blocks, with some blocks masked, and the remaining blocks reassembled into a new sample, denoted as $x_i^1$. The origi-

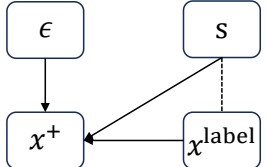

Figure 1: The SCM for Equation (1).

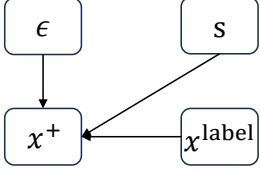

Figure 2: The SCM for $p^{\text{PI}}(x^+, x^{\text{label}}, s)$.

nal sample is then referred to as $x_i^2$. Thus, the augmented dataset in SSL (whether D-SSL or G-SSL) is represented as $X_{tr}^{aug} = \{x_i^1, x_i^2\}_{i=1}^N$. The pair $\{x_i^1, x_i^2\}$ forms the $i$-th pair, and SSL aims to learn a feature extractor $f$ from these pairs.

The objective of D-SSL methods typically consists of two components: alignment and regularization (Wang & Isola, 2020; Chen et al., 2021a). The alignment part is to maximize the similarity between samples that share the same pair in the embedding space, and the regularization part aims to constrain the learning behavior via inductive bias, e.g., SimCLR (Chen et al., 2020) constrains the feature distribution to satisfy a uniform distribution. Meanwhile, G-SSL methods (He et al., 2022) can be regarded as implementing alignment of samples within a pair based on an encoding-decoding structure, by inputting sample $x_i^1$ into this structure to generate a sample, and making it as consistent as possible with sample $x_i^2$. It is noteworthy that "alignment" in D-SSL is often implemented based on anchor points, that is, viewing one sample in a pair as an anchor, the training process of such SSL methods can be seen as gradually pulling the other sample in this pair towards the anchor. The concept of anchor is also applicable to G-SSL, where $x_i^2$ is viewed as the anchor, and thus the training process of such SSL methods can be viewed as gradually constraining $x_i^1$ to approach $x_i^2$.

Regardless of whether it is G-SSL or D-SSL, the anchor can be regarded as a learning target. Specifically, SSL can be interpreted as follows: In a data augmentation pair, one sample (the anchor) is designated as the target. By constraining the other augmented sample in the feature space to move toward this anchor, consistency in feature representations is achieved. This dynamic adjustment causes samples within the same pair to become tightly clustered, thereby forming an effect similar to a local cluster center. In traditional classification problems, the common approach is to first project samples into a label space and then constrain them to move toward their corresponding one-hot labels to achieve supervision. In contrast, SSL directly applies constraints in the feature space, which means that the anchor effectively takes on the role of a "label." In this unsupervised setting, the anchor provides a supervisory signal similar to that of a label. Therefore, it can be argued that labels manifest differently across various scenarios, and in SSL scenarios, the anchor represents this "implicit label".

Based on the above discussion, when we consider the anchor

as the label or the center of clustering, each mini-batch in the training phase can be viewed as a multi-class classification task. Specifically, $X_{tr}^{aug} = \{x_i^+, x_i^{\text{anchor}}\}_{i=1}^N$ consists of data from $N$ categories, where $x_i^+$ is the positive sample of the $i$-th category whose clustering center is $x_i^{\text{anchor}}$. Furthermore, the variability of data across mini-batches implies that each mini-batch corresponds to a distinct training task or domain.

## 3. Motivation and Causal Analysis

In this section, we first offer a plausible explanation for the OOD generalization capability of SSL models from a task distribution perspective (the training tasks and the test task is different, this is the key concept of the OOD generalization). Next, based on causal inference, we demonstrate that 1) during the training phase, SSL exploits spurious correlations to learn certain tasks, thereby degrading its OOD generalization performance; 2) the tasks addressed in SSL training are difficult to model with a single Structural Causal Model, thus, it is challenging to eliminate the confounding factors involved in learning different tasks through one unified method. Finally, through theoretical analysis, we present that even in the case of spurious associations, we can further improve the OOD generalization of SSL by constraining the data distribution.

### 3.1. Formation of the Problem: Causal Perspective

According to Section 2, we can infer that different mini-batches correspond to distinct classification tasks. Therefore, the training process of SSL can be described as follows: given a distribution over tasks and the corresponding data distribution of each task (refer to **Appendix E** for more details about the definitions of task distribution and data distribution), the SSL model can be regarded as be learned based on various training tasks and their corresponding data. The performance of the SSL model is then evaluated on test tasks that are disjoint from the training tasks. This learning paradigm can be regarded as estimating the true task distribution from discrete training tasks, enabling the SSL model to generalize to new, unseen tasks (i.e., test tasks). This also explains well why the SSL model exhibits good performance in transfer tasks (Chen et al., 2020; Grill et al., 2020a; Zbontar et al., 2021), i.e., it has good OOD generalization (refer to **Appendix E** for more details about the relation between OOD generalization and generalization on task distributions). However, machine learning models are prone to learning spurious correlations during the training phase (Wang et al., 2023a; 2022). For example, compared to the foreground features of input data, researchers have found that machine learning models tend to rely on the superficial texture information or background information of the data for decision-making (Geirhos et al., 2018; Qiang et al., 2022; Xu et al., 2020). Therefore, although the SSL

models have been effective in OOD generalization, it may still face the challenge of spurious correlations.

We further analyze the above challenge from the perspective of data generation and causal inference. Without loss of generality, for each pair in the SSL training process, we denote the anchor as $x^{\text{label}}$ and the other sample as $x^+$. Based on (Zimmermann et al., 2021; Von Kügelgen et al., 2021), $x^+$ can be regarded as caused by anchor $x^{\text{label}}$, an unobserved latent variable $s \in \mathbb{R}^n$ and an independent noise variable $\epsilon$ with the following formulation:

$$x^+ = F(s, x^{\text{label}}) + \epsilon, \tag{1}$$

where $F$ is a reversible injective function. From a causal perspective, Equation (1) can be reformulated as the Structural Causal Model (SCM) shown in Figure 1. The solid arrow indicates that there is a direct causal relationship between the two variables, e.g., $x^{\text{label}} \rightarrow x^+$ states that $x^{\text{label}}$ is the direct cause of obtaining $x^+$. The dotted line indicates that the relationship between the variables is not clear and varies with different environments. Notably, this paper focuses exclusively on scenarios where the semantic information within $x^+$ is related only to $x^{\text{label}}$, that is, $s$ does not contain any causal semantics related to the task. Next, we examine two examples illustrated in Figure 3. In Figure 3 (a), $s$ represents the assigned color, for example, the color of numbers varies by category, as in the ColoredMNIST dataset (Arjovsky et al., 2019). Here, $e_{\text{id}}$ denotes the class index. Consequently, within a mini-batch during training, samples from different classes may have a different texture color. In Figure 3 (b), $s$ indicates assigned stylistic attributes, e.g., sketches, cartoon styles, or photographs, and $e_{\text{id}}$ denotes the batch index. This scenario commonly occurs in multi-view or domain generalization contexts, like the tasks in the PACS dataset (Li et al., 2017). Therefore, during training, different batches may exhibit different styles, with samples under each style possessing unique appearance attributes. In both figures, $s$ does not capture the foreground semantics between $x^{\text{label}}$ and $x^+$, and the correlation between $x^{\text{label}}$ and $x^+$ may vary depending on the settings.

Based on Figure 3 (a) and (b), we obtain that the causal relationship between $x^{\text{label}}$ and $s$ changes with unknown environmental variations, making it difficult to eliminate based on a unified causal criterion proposed in (Pearl et al.). From Figure 3 (a), due to the existence of path "$x^{\text{label}} \cdots s \rightarrow x^+$", the following proposition states that the correlation between $x^{\text{label}}$ and $x^+$ is influenced by $s$.

**Proposition 3.1.** *Revisiting SSL from a pairwise perspective and assuming that the two samples in each pair satisfy Equation (1), we can obtain that the learned SSL model will use non-causal factor, i.e., the unobserved latent variable s, to measure the similarity or reconstruct in a pair.*

Detailed proof of **Proposition** 3.1 is provided in **Appendix**

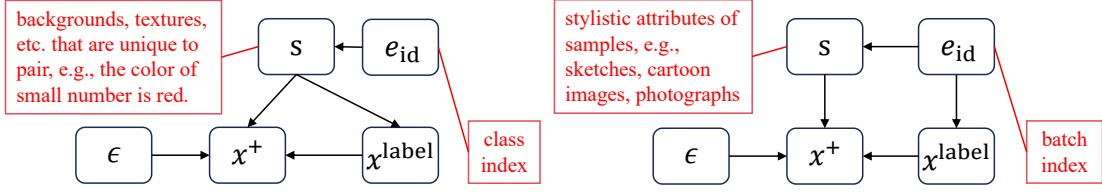

(a) Example task related to ColoredMNIST dataset     (b) Example task related to PACS dataset

Figure 3: Two specific instances illustrate the variability in the causal relationship between $x^{\text{label}}$ and $s$ due to environmental changes. The black squares are variables and the arrows indicate causality.

A.1. Notably, when SSL models measure the similarity or reconstruct between paired elements using non-causal factors and we can not use one unified method to eliminate the non-causal factors for all related tasks, the extracted representations may incorporate semantics irrelevant to the task. From the pairwise perspective, this may result in SSL failing to effectively learn each specific task, thereby hindering the modeling of the task distribution and ultimately reducing the OOD generalization ability of SSL.

### 3.2. Motivation: Post-Intervention Distribution

As shown in Figure 3, regardless of the correlation between $s$ and $x^{\text{label}}$, the generation mechanism of $x^+$ is invariant. Because SCMs can be considered as a joint probability distribution, we use the following distribution set to represent the joint probability distribution related to Figure 1:

$$\mathcal{D} = \left\{ p(x^+, x^{\text{label}}, s) = p(x^+|x^{\text{label}}, s)p(x^{\text{label}})p(s|x^{\text{label}}) \middle| p(x^{\text{label}}), p(s|x^{\text{label}}) > 0 \right\}. \quad (2)$$

Instead of exploring what the specific structure of "$x^{\text{label}} \cdots \cdot s \to x^+$", we propose to consider using Post-Intervention Distribution (PID) to model $p(x^+, x^{\text{label}}, s)$, defined as:

**Definition 3.2.** If the distribution $p(x^+, x^{\text{label}}, s) = p(x^+|x^{\text{label}}, s)p(x^{\text{label}})p(s)$, then $p(x^+, x^{\text{label}}, s)$ is defined as PID. $x^{\text{label}}$ and $s$ are independent in PID.

We use $p^{\text{PI}}$ to denote distributions belonging to the PID family. As we can see, $p(x^+|x^{\text{label}}, s)$ is both a component of $p^{\text{PI}}(x^+, x^{\text{label}}, s)$ and a result of the unchanged causal mechanism "$s \to x^+ \leftarrow x^{\text{label}}$" in Figure 1. Then, the SCM of $p^{\text{PI}}(x^+, x^{\text{label}}, s)$ is shown as Figure 2. In this new distribution, because there are no paths between $s$ and $x^{\text{label}}$, we can obtain that $x^+$ and $x^{\text{label}}$ are only correlated through the stable causal relation $x^+ \leftarrow x^{\text{label}}$. Then, from a probabilistic perspective, what we argue is that compared to SSL models trained on batches satisfying other distribution constraints in $\mathcal{D}$, SSL models trained on batches that meet the PID distribution constraint have the lowest worst-case risk. To support this, we build upon (Pearl, 2009) by introducing an assumption regarding the invertibility of functions:

**Assumption 3.3.** There exist functions $F_{x^{\text{label}}}$, $F_s$, and noise variables $\epsilon_{x^{\text{label}}}$, $\epsilon_s$, such that the pair $(x^{\text{label}}, s)$ can

be recovered from the latent variable $x^+$ via the inverse mapping: $(x^{\text{label}}, s) = F^{-1}(x^+ - \epsilon) = \left( F_{x^{\text{label}}}(x^+ - \epsilon_{x^{\text{label}}}), F_s(x^+ - \epsilon_s) \right)$, where $\epsilon$ denotes the combined noise introduced in the transformation process.

**Assumption** 3.3 implies that $x^{\text{label}} \perp\!\!\!\perp_{\text{PI}} s|x^+$, and the intuitive explanation of **Assumption** 3.3 can be found in **Appendix** F. Based on Section 2 and Section 3.1, both D-SSL and G-SSL share a common objective: aligning the positive sample in a pair with its corresponding anchor. Thus, the learning objectives of D-SSL and G-SSL can be unified as maximizing $p_f(x^{\text{label}}|x^+)$. The difference lies in how they achieve $p_f(x^{\text{label}}|x^+)$. For example, the training data is first projected to the feature space by $f$, then SimCLR uses a contrastive loss to achieve $p_f(x^{\text{label}}|x^+)$, while MAE employs the $L_2$-norm achieve $p_f(x^{\text{label}}|x^+)$. We obtain:

**Theorem 3.4.** *From a Bayesian perspective, the alignment component of the SSL objective, i.e., encouraging paired samples to be similar in the feature space, can be interpreted as maximizing the conditional likelihood* $p_f(x^{\text{label}} \mid x^+)$. *Given a model $f$, the risk over a mini-batch drawn from environment $e \in \mathcal{D}$, treated as a distributional constraint, can be defined as:* $\mathcal{L}^e(f) = \mathbb{E}_{p^e(x^+, x^{\text{label}})} \left[ -\log p_f(x^{\text{label}} \mid x^+) \right]$, *where $p^e(x^+, x^{\text{label}})$ denotes the joint distribution in environment $e$. Under **Assumption** 3.3, if $f^* = \arg\min \mathcal{L}^e(f)$ for all $e \in PID$, then $f^*$ is minimax optimal across all environments in $\mathcal{D}$, that is, $f^* = \arg\min_f \max_{e \in \mathcal{D}} \mathcal{L}^e(p_f(x^{\text{label}} \mid x^+))$.*

Detailed proof of **Theorem** 3.4 is provided in **Appendix** A.2. **Theorem** 3.4 implies that when $\mathcal{D}$ is sufficiently large and diverse, no other $f$ obtained from training on any distribution can achieve better worst-case OOD performance than the PID (refer to **Appendix** H for more detail about why the worst-case OOD performance is effective). Notably, transferring Figure 1 to Figure 2 is similar to backdoor adjustment in causal inference (Pearl et al.). However, from backdoor adjustment pespective, it is straightforward to explain why PID can improve the OOD performance of D-SSL: during the learning of each task, PID eliminates the influence of background semantic confounding (Qiang et al., 2022). However, for G-SSL, regardless of the relationship between $s$ and $x^{\text{label}}$, due to the generation purpose, G-SSL

inherently requires encoding background semantics. Thus, explaining the improvement of OOD performance of G-SSL from the backdoor adjustment perspective is incorrect. Therefore, **Theorem** 3.4 is provided to explain why PID can improve the OOD performance of both D-SSL and G-SSL simultaneously. Also, an intuitive explanation of **Theorem** 3.4 is shown in **Appendix** F. Moreover, **Theorem** 3.4 motivates us to design a new mini-batch sampling strategy to ensure that the resulting mini-batches satisfy PID, thereby improving the OOD generalization of SSL.

## 4. The Proposed Method

In this section, we present the proposed method which consists of two stages. In the first stage, we use a regularized latent variable model (RLVM), e.g., variational autoencoder (VAE) (Kingma & Welling, 2013b), to learn the underlying distribution $p(x^+, x^{\mathrm{label}}, s)$ for each batch task. In the second stage, based on commonly used instruments in causal effect estimation (Rosenbaum & Rubin, 1981; King & Nielsen, 2019; Austin, 2011), we first introduce a way to calculate the propensity score. Then, we use the learned distribution and propensity score to obtain a sampling strategy that can create a PID based on training data.

### 4.1. Learning Regularized Latent Variable Model

As shown in Equation (2), to learn the underlying joint distribution $p(x^+, x^{\mathrm{label}}, s)$ for each batch task, we need to know $p(x^+|x^{\mathrm{label}}, s), p(x^{\mathrm{label}}), p(s|x^{\mathrm{label}})$ in each batch task. Because that $p(x^+|x^{\mathrm{label}}, s)$ is the unchanged causal mechanism, so we can use a unified f to model $p(x^+|x^{\mathrm{label}}, s)$ in all tasks. Based on the discussion in Section 2, we obtain that $x^{\mathrm{label}}$ is regarded as the label. So, $p(x^{\mathrm{label}})$ can be regarded as the label distribution, and we can represent it with the same uniform distribution in all tasks. Based on the mean-field approximation (Blei et al., 2017; Sriperumbudur et al., 2013) which can be expressed as a closed form of the true prior, we obtain that when the causal relationship between the latent covariate and the label changes with the tasks, an exponential family distribution has the ability to model the conditional distribution $p(s|x^{\mathrm{label}})$, thus, we have the following assumption for each batch task:

**Assumption 4.1.** Denote the mini-batch task index as $e$, the correlation between $x^{\mathrm{label}}$ and $s$ in the data distribution $p^e(x^+, x^{\mathrm{label}}, s)$ of a task can be characterized by:

$$p_{\mathrm{T},\lambda^e}^e(s|x^{\mathrm{label}}) = \prod_{i=1}^n \frac{Q_i(s_i)}{K_i^e(x^{\mathrm{label}})} \exp[\sum_{j=1}^k T_{ij}(s_i)\lambda_{ij}^e(x^{\mathrm{label}})], \quad (3)$$

where $n$ is the dimension of the latent variable $s$, $k$ is the dimension of each sufficient statistic, $s_i$ is the $i$-th element of $s$, $Q = [Q_i]: s \to \mathbb{R}^n$ is the base measure, $T = [T_{ij}]: s \to \mathbb{R}^{nk}$ is the sufficient statistics, $K^e = [K_i^e]: x^{\mathrm{label}} \to \mathbb{R}^n$ is the normalizing constraint, and $\lambda^e = [\lambda_{ij}^e]: x^{\mathrm{label}} \to \mathbb{R}^{nk}$.

Note that $k$, Q, and T are determined by the type of chosen exponential family distribution and thus independent of $e$, this guides us to constrain all batch tasks to share these parameters during the training phase. For ease of calculation, we set $Q_i(\cdot) = \exp(\cdot/-2)$ and $K^e$ as the feature normalization operator. For $\lambda^e$, since it varies with $e$, we implement it as the output of a network. Specifically, we first average all the data of a batch, then feed it into a learnable network $g$, and output the corresponding $\lambda^e$. For T, we need to guarantee it to be a sufficient statistic, one simple way to implement this is the constant transformation. Considering the identifiability of the parameters, we implement it as $T_{ij}(\cdot) = a_{ij} \times \cdot$, where $A = [a_{ij}]$ is a learnable parameter. Up to this point, we obtain the implementation of $p_{\mathrm{T},\lambda^e}^e(s|x^{\mathrm{label}})$ as $p_{g,A}(s|x^{\mathrm{label}})$. Then, we implement the conditional generative model in each $e \in \mathcal{D}$ with parameters $\theta = (\mathrm{f}, g, A)$ as: $p_\theta^e(x^+, s|x^{\mathrm{label}}) = p_{\mathrm{f}}(x^+|s, x^{\mathrm{label}})p_{g,A}(s|x^{\mathrm{label}})$.

Motivated by the VAE, we estimate the above conditional generative model with the following regularized evidence lower bound (ELBO) in each batch distribution $e$:

$$\mathcal{L}_{\theta,\phi}^e = \mathbb{E}_{q_\phi(s|x^+,x^{\mathrm{label}})}[\log p_{\mathrm{f}}(x^+|s, x^{\mathrm{label}})]$$
$$-\mathrm{KL}(q_\phi(s|x^+, x^{\mathrm{label}}) \, \| \, p_{g,A}(s|x^{\mathrm{label}})) - \alpha \underbrace{\sum_{i,j} A_{\cdot,i} \cdot A_{\cdot,j}}_{\text{Regularizer}}, \quad (4)$$

where $A_{\cdot,i}$ is the column vector of A, $\mathrm{KL}(\cdot)$ is the KL-divergence, and $\alpha$ is a hyperparameter. As for $q_\phi(s|x^+, x^{\mathrm{label}})$, it is implemented by a learnable network $\phi$ that outputs the mean and variance, and we use reparameterization trick (Kingma & Welling, 2013a) to deal with it during training. The last term of Equation (4) is to constrain the column vector orthogonality of A. The training process of Equation (4) is similar to meta-learning, e.g., Prototype Networks (Snell et al., 2017), because we construct a series of tasks in the training phase. Thus, from a meta-learning perspective, training with Equation (4) also indicates that the learned $\theta$ can be adaptable for all available tasks.

We further show that we can uniquely recover the model parameter $\theta$ up to an equivalence relation. Specifically, we first give the definition of the equivalence relation based on (Motiian et al., 2017):

**Definition 4.2.** $(\mathrm{f}, g, A) \sim_W (\mathrm{f}', g', A')$, if and ony if there exists an invertible matrix $W \in \mathbb{R}^{nk \times nk}$ and a vector $b \in \mathbb{R}^{nk}$, $A(\mathrm{f}^{-1}(x)) = WA'(\mathrm{f}'^{-1}(x)) + b, \forall x \in X_{tr}^{aug}$.

Then, motivated by (Khemakhem et al., 2020), the identifiability condition of $\theta$ can be presented as:

**Theorem 4.3.** *Suppose that* $p_\theta^e(x^+, s|x^{\mathrm{label}}) = p_{\mathrm{f}}(x^+|s, x^{\mathrm{label}})p_{g,A}(s|x^{\mathrm{label}})$ *and the generation process of* $X^+$ *can be represented by the SCM depicted in Figure 1, **a sufficient condition** for* $\theta = (\mathrm{f}, g, A)$ *to be* $\sim_A$*-identifiable is given as: 1) Suppose that*

$p_\epsilon(x^+ - \mathrm{f}(x^{\mathrm{label}}, s)) = p_{\mathrm{f}}(x^+|x^{\mathrm{label}}, s)$, $\phi_\varepsilon$ *is the characteristic function of* $p_\epsilon(x^+ - \mathrm{f}(x^{\mathrm{label}}, s))$, *and the set* $\{x^+|\phi_\varepsilon(x^+) = 0\}$ *has measure zero; 2) The sufficient statistics* $\mathrm{T}$ *are differentiable almost everywhere, and* $[T_{ij}]_{1 \le j \le k}$ *are linearly independent on any subset of* $X^+$ *with measure greater than zero; 3) There exist* $nk + 1$ *distinct pairs* $(x_0^{\mathrm{label}}, e_0), \cdots, (x_n^{\mathrm{label}}k, e_{nk})$ *such that the* $nk \times nk$ *matrix* $\mathrm{L} = (\lambda^{e_1}(x_1^{\mathrm{label}}) - \lambda^{e_0}(x_0^{\mathrm{label}}), \cdots, \lambda^{e_{nk}}(x_{nk}^{\mathrm{label}}) - \lambda^{e_0}(x_0^{\mathrm{label}}))$ *is invertible.*

Detailed proof of **Theorem** 4.3 is provided in **Appendix** A.3. In Equation (4), we constrain the column vector orthogonality of A, this can lead to the linearly independence of elements of T, thus, the second assumption of **Theorem** 4.3 holds. Meanwhile, according to Section 2, we can obtain that each ancestor training sample can be regarded as a class, by combining different classes with each other, we can construct adequate tasks, thus, the third assumption of **Theorem** 4.3 can easily holds. Therefore, based on **Theorem** 4.3, we can obtain that $\theta$ can be uniquely recovered. Moreover, we do not theoretically prove that the latent variable model can directly identify the spurious variable $s$. In this paper, the identification of $s$ is based on **Assumption** 4.1. **Theorem** 4.3 is also based on **Assumption** 4.1. The key result of **Theorem** 4.3 is that we can uniquely identify $\phi$ and $(\mathrm{f}, g, \mathrm{A})$, thus, $s$ is conditionally identified (the detailed explanation of the identifiability of spurious variable $s$ is provided in **Appendix** G).

According to **Assumption** 3.3, both $x_i^{\mathrm{label}}$ and $s$ can be obtained through an invertible neural network. Training such an invertible neural network typically requires training data in the form of $\{(x_i^{\mathrm{label}}, s_i, x_i^+)\}_{i=1}^N$. However, we do not adopt this mechanism directly because we do not have access to such training data. In particular, we are unable to provide the corresponding $s_i$ for each pair. Therefore, we opt to use a RLVM approach instead.

### 4.2. The Proposed Mini-Batch Sampling Strategy

As shown in (Rosenbaum & Rubin, 1981), balancing score matching has become a useful tool in the average treatment effect estimation. One of its purposes is to reveal the true causal relationship from the observational data, defined as:

**Definition 4.4.** A balancing score $ba(s)$ is a function of covariate $s$ that satisfies: $s \perp\!\!\!\perp x^{\mathrm{label}}|ba(s)$.

From (Rosenbaum & Rubin, 1981), we can obtain that many functions can be used as a balancing score, among them, propensity score $p(x^{\mathrm{label}}|s)$ is the coarsest one. Motivated by this, given the batch task with "nu" pairs, we define the propensity score under the SSL scenario as:

**Definition 4.5.** The propensity score for a batch task in SSL scenario is $mi(s) = [p(x_j^{\mathrm{label}}|s)]_{j=1}^{\mathrm{nu}}$.

Then, given a function $ba(s)$, we present a sufficient condition that it can be the balancing score:

**Corollary 4.6.** *Let* $ba(s)$ *be a function of* $s$, ***a sufficient condition*** *that* $ba(s)$ *can be regarded as a balancing score is that there exists a function* $\psi$ *such that* $mi(s) = \psi(ba(s))$.

The proof of **Corollary** 4.6 can be obtained based on **Theorem** 1 and **Theorem** 2 in (Rosenbaum & Rubin, 1981). We use $ba^e(s)$ to denote the balancing score for a specific batch task $e$ of SSL. Then, the corresponding propensity score can be represented as $mi^e(s) = [p^e(x_j^{\mathrm{label}}|s)]_{j=1}^{\mathrm{nu}}$, which can be derived from $p_{\mathrm{T},\lambda^e}^e(s|x^{\mathrm{label}})$ as defined in Equation (3):

$$p^e(x_j^{\mathrm{label}}|s) = \frac{p_{g,\mathrm{A}}(s|x_j^{\mathrm{label}})p^e(x_j^{\mathrm{label}})}{\sum_{j=1}^{\mathrm{nu}} p_{g,\mathrm{A}}(s|x_j^{\mathrm{label}})p^e(x_j^{\mathrm{label}})}, \quad (5)$$

where $p^e(x_j^{\mathrm{label}}) = 1/\mathrm{nu}$, because that $p^e(x_j^{\mathrm{label}})$ is defined empirically as a uniform distribution.

Based on **Corollary** 4.6, we set $\psi$ as identical transformation and propose to use the propensity score computed from Equation (5) directly as our balancing score, e.g., $ba(s) = mi^e(s)$. Next, we derive the proposed sampling strategy. When given the training data $X^{tr} = \{x_i^+, x_i^{\mathrm{label}}\}_{i=1}^{\mathrm{mu}}$ with "mu" pairs, we can obtain $\lambda^e$ of Equation (5) based on the mean of the entire dataset. Then, for each pair, we firstly take one $s$ based on the learned $q_\phi^e(s|x^+, x^{\mathrm{label}})$ and secondly obtain $ba(s)$ by setting $\mathrm{nu} = \mathrm{mu}$ in Equation (5) (please refer to the **Step 3** in **Appendix** G for the explanation of the spatial structure of $s$ and the rationale for sampling $s$ from $q_\phi^e(s|x^+, x^{\mathrm{label}})$). Finally, we match $ba(s)$ of the selected pair with $1 \le a \le N - 1$ different pairs that have the same/closest balancing score. The detailed sampling strategy is shown in **Algorithm** 1. Meanwhile, we set the distance metric $d(\cdot)$ in **Algorithm** 1 as the measurement of distribution distance, e.g., JS-divergence (Endres & Schindelin, 2003), because that the propensity score itself can be seen as a distribution. Denote the distribution obtained from **Algorithm** 1 as $\hat{p}(x^+, x^{\mathrm{label}}, s)$, then we have:

**Theorem 4.7.** *If* $d(ba(s_j), ba(s_i)) = 0$ *in **Algorithm** 1, the obtained mini-batch is regarded as sampling from a PID, e.g.,* $\hat{p}(x^+, x^{\mathrm{label}}, s) = p^{\mathrm{PI}}(x^+, x^{\mathrm{label}}, s)$.

Detailed proof and high-level explanation of **Theorem** 4.7 is provided in **Appendix** A.4 and F. Based on **Theorem** 4.7, if at each step, we achieve perfect matching (i.e., $ba(s_j) = ba(s_i)$), and the obtained mini-batch samples can be regarded as sampled from the PID. It is worth noting that concepts such as the balancing score, propensity score, and Equation (5) are not novel to this work, because they have been extensively studied by previous researchers (Rosenbaum & Rubin, 1981; King & Nielsen, 2019; Austin, 2011). However, what we aim to highlight is that the novelty of our approach lies in the proposed sampling strategy, e.g.,

---

**Algorithm 1** Proposed Mini-Batch Sampling Strategy

---

**Input:** Training dataset $X^{tr} = \{x_i^+, x_i^{\text{label}}\}_{i=1}^{mu}$, balancing score function $ba(\cdot)$, distance metric $d : ba(\cdot) \times ba(\cdot) \to \mathbb{R}$
**Output:** Mini-batch data $D^{\text{PI}}$ consisting of $a + 1$ examples

 1: $D^{\text{PI}} \leftarrow \emptyset; i \leftarrow 0$
 2: **while** $i = 0$ **do**
 3:    Randomly sample $(x_i^+, x_i^{\text{label}})$ from $X_{aug}^{tr}$, add to $D^{\text{PI}}$; Compute $ba(s_i)$ from $(x_i^+, x_i^{\text{label}})$
 4:    $i \leftarrow i + 1$
 5: **end while**
 6: **for** $1 \le i \le a$ **do**
 7:    $j \leftarrow \arg\min_{x_j^+ \in X_{aug}^{tr} \setminus D^{\text{PI}}} d(ba(s_j), ba(s_i))$
 8:    Add $(x_j^+, x_j^{\text{label}})$ to $D^{\text{PI}}$
 9:    $i \leftarrow i + 1$
10: **end for**

---

Algorithm 1, which enables direct sampling of a mini-batch that satisfies the post-intervention distribution under perfect matching conditions, i.e., when $d(ba(s_j), ba(s_i)) = 0$.

As shown in Section 4.2 and Section 4.1, the proposed method has two main phases with the following complexity analysis per mini-batch (mini-batch size $B$, dataset size $D$): **1) Regularized Latent Variable Model Training**: (a) $q_\phi(s|x^+, x^{\text{label}})$: Each sample requires a forward pass with cost $O(C_\phi)$, totaling $O(B \cdot C_\phi)$. (b) $p_f(x^+|s, x^{\text{label}})$: Each sample incurs a cost $O(C_f)$, totaling $O(B \cdot C_f)$. (c) $g$ for $\lambda^e$: Computed once per mini-batch with cost $O(C_g)$. (d) KL-Divergence: Involves operations over the latent dimension $n$ and sufficient statistic dimension $k$, contributing $O(B \cdot n \cdot k)$. (e) Orthogonality Regularization: Requires $O(n \cdot k^2)$, which is constant when $n$ and $k$ are small. Thus, the training phase complexity is approximately: $O(B \cdot (C_\phi + C_f + n \cdot k) + C_g + n \cdot k^2)$; **2) Algorithm 1**: (a) Propensity Score Calculation: For each sample, computing scores across the $D$ candidates costs $O(D \cdot n \cdot k)$, leading to a total of $O(D^2 \cdot n \cdot k)$ for the mini-batch. (b) Matching Operation: A brute-force matching over $D$ samples yields an additional $O(D^2)$. Therefore, the sampling phase has an overall complexity of approximately: $O(D^2 \cdot n \cdot k)$; **3) Overall Complexity**: The combined complexity per mini-batch is: $O(B \cdot (C_\phi + C_f + n \cdot k) + C_g + n \cdot k^2 + D^2 \cdot n \cdot k)$. The symbols $C_\phi$, $C_f$, and $C_g$ represent the computational cost for a single forward pass (or operation) of each respective network module.

# 5. Experiments

In this section, we first introduce the datasets used in experiments. Next, we evaluate our method[1] on multiple tasks, including unsupervised learning, semi-supervised learning,

---

[1]Codes of the proposed Sampling Strategy can be found in https://github.com/ML-TASA/PID-SSL

transfer learning, and few-shot learning. We introduce the experimental setups in the corresponding sections. Finally, we perform ablation studies. All results reported are the averages of five runs performed on NVIDIA RTX 4090 GPUs. More experiments are shown in **Appendix C**.

## 5.1. Benchmark Datasets

For unsupervised learning, we select ImageNet-100 (Tian et al., 2020) and ImageNet (Deng et al., 2009). For semi-supervised learning, we select ImageNet (Deng et al., 2009) for evaluation. For transfer learning, we select PASCAL VOC (Everingham et al., 2010) and COCO (Lin et al., 2014). For few-shot learning, we evaluate the proposed method on Omniglot (Lake et al., 2019), miniImageNet (Vinyals et al., 2016), and CIFAR-FS (Bertinetto et al., 2018).

## 5.2. Empirical Analysis

In this article, we primarily address the OOD generalization of SSL. Our experimental design consists of the following steps: First, we validate that the proposed sampling strategy enhances the performance of SSL methods in in-distribution scenarios using unsupervised tasks. Second, we classify OOD tasks by difficulty into semi-supervised tasks, transfer learning tasks, and few-shot learning tasks, and subsequently evaluate the proposed sampling strategy on these tasks. Meanwhile, we conduct experiments on generative SSL, the evaluation is provided in **Appendix C.1**.

**Experimental setup.** Our proposed sampling strategy is compatible with any D-SSL or G-SSL model. In the standard training procedure of SSL, a mini-batch is randomly sampled from the training data before each iteration. In contrast, our method replaces this random sampling step with a structured mini-batch construction process defined by **Algorithm 1**. Specifically, our approach integrates seamlessly into existing SSL frameworks by substituting the mini-batch sampling component with **Algorithm 1**, while leaving all other aspects of the SSL training pipeline unchanged. As a result, the overall training procedure and hyperparameter settings remain identical to those used in the baseline methods. Therefore, for all our experiments, we retain the original hyperparameter configurations to ensure a fair and consistent comparison.

**Results on unsupervised learning tasks.** **Table 1** shows the top-1 and top-5 linear classification accuracies on ImageNet-100 and ImageNet for unsupervised learning task. We can observe that applying the proposed method achieves stable performance improvement, and significantly outperforms the state-of-the-art (SOTA) methods on all datasets.

**Results on semi-supervised learning tasks.** **Table 2** shows the results on ImageNet for semi-supervised learning task. No matter 1% or 10% of the labels are available in 1000

Table 1: The Top-1 and Top-5 classification accuracies of linear classifier on the ImageNet-100 dataset and the Top-1 results for ImageNet dataset with ResNet-50.

| Method | ImageNet-100 | | ImageNet | |
|---|---|---|---|---|
| | Top-1 | Top-5 | 400 Epochs | 1000 Epochs |
| SimCLR (Chen et al., 2020) | $70.15 \pm 0.16$ | $89.75 \pm 0.14$ | $69.24 \pm 0.21$ | $70.45 \pm 0.30$ |
| MoCo (He et al., 2020) | $72.80 \pm 0.12$ | $91.64 \pm 0.11$ | $69.76 \pm 0.14$ | $71.16 \pm 0.23$ |
| SimSiam (Chen & He, 2021) | $73.01 \pm 0.21$ | $92.61 \pm 0.27$ | $70.86 \pm 0.34$ | $71.37 \pm 0.22$ |
| Barlow Twins (Zbontar et al., 2021) | $75.97 \pm 0.23$ | $92.91 \pm 0.19$ | $70.22 \pm 0.15$ | $73.29 \pm 0.13$ |
| SwAV (Caron et al., 2020) | $75.78 \pm 0.16$ | $92.86 \pm 0.15$ | $70.78 \pm 0.34$ | $75.32 \pm 0.11$ |
| DINO (Caron et al., 2021) | $75.43 \pm 0.18$ | $93.32 \pm 0.19$ | $71.98 \pm 0.26$ | $73.94 \pm 0.29$ |
| RELIC v2 (Tomasev et al., 2022) | $75.88 \pm 0.15$ | $93.52 \pm 0.13$ | $71.84 \pm 0.21$ | $72.17 \pm 0.20$ |
| VICRegL (Bardes et al., 2022) | $75.96 \pm 0.19$ | $92.97 \pm 0.26$ | $72.14 \pm 0.20$ | $75.07 \pm 0.23$ |
| SimCLR + Ours | $73.32 \pm 0.15$ | $91.74 \pm 0.18$ | $72.24 \pm 0.20$ | $73.66 \pm 0.25$ |
| MoCo + Ours | $74.71 \pm 0.22$ | $93.89 \pm 0.17$ | $72.04 \pm 0.21$ | $74.06 \pm 0.20$ |
| SimSiam + Ours | $75.66 \pm 0.18$ | $95.02 \pm 0.21$ | $72.96 \pm 0.22$ | $73.67 \pm 0.21$ |
| Barlow Twins + Ours | $77.77 \pm 0.18$ | $94.99 \pm 0.20$ | $73.08 \pm 0.21$ | $75.89 \pm 0.17$ |
| SwAV + Ours | $76.99 \pm 0.11$ | $95.03 \pm 0.20$ | $73.25 \pm 0.24$ | $77.42 \pm 0.21$ |
| DINO + Ours | $77.47 \pm 0.15$ | $\mathbf{96.01} \pm 0.17$ | $74.21 \pm 0.20$ | $75.99 \pm 0.17$ |
| VICRegL + Ours | $\mathbf{78.20} \pm 0.14$ | $95.07 \pm 0.21$ | $\mathbf{74.91} \pm 0.14$ | $\mathbf{77.77} \pm 0.21$ |

Table 2: The semi-supervised learning accuracies ($\pm 95\%$ confidence interval) on the ImageNet dataset with the ResNet-50 pre-trained on the ImageNet dataset.

| Method | Epochs | 1% | | 10% | |
|---|---|---|---|---|---|
| | | Top-1 | Top-5 | Top-1 | Top-5 |
| MoCo (He et al., 2020) | 200 | $43.8 \pm 0.2$ | $72.3 \pm 0.1$ | $61.9 \pm 0.1$ | $84.6 \pm 0.2$ |
| BYOL (Grill et al., 2020b) | 200 | $54.8 \pm 0.2$ | $78.8 \pm 0.1$ | $68.0 \pm 0.2$ | $88.5 \pm 0.2$ |
| BYOL + Ours | 200 | $46.5 \pm 0.2$ | $74.4 \pm 0.2$ | $63.6 \pm 0.3$ | $85.6 \pm 0.2$ |
| MoCo + Ours | 200 | $\mathbf{57.4} \pm 0.2$ | $\mathbf{80.1} \pm 0.2$ | $\mathbf{71.4} \pm 0.2$ | $\mathbf{90.2} \pm 0.1$ |
| SimCLR (Chen et al., 2020) | 1000 | $48.3 \pm 0.2$ | $75.5 \pm 0.1$ | $65.6 \pm 0.1$ | $87.8 \pm 0.2$ |
| MoCo (He et al., 2020) | 1000 | $52.3 \pm 0.1$ | $77.9 \pm 0.2$ | $68.4 \pm 0.1$ | $88.0 \pm 0.2$ |
| BYOL (Grill et al., 2020b) | 1000 | $56.3 \pm 0.2$ | $79.6 \pm 0.2$ | $69.7 \pm 0.2$ | $89.3 \pm 0.1$ |
| Barlow Twins (Zbontar et al., 2021) | 1000 | $55.0 \pm 0.1$ | $79.2 \pm 0.1$ | $67.7 \pm 0.2$ | $89.3 \pm 0.2$ |
| RELIC v2 (Tomasev et al., 2022) | 1000 | $55.2 \pm 0.2$ | $80.0 \pm 0.1$ | $68.0 \pm 0.2$ | $88.9 \pm 0.2$ |
| VICRegL (Bardes et al., 2022) | 1000 | $54.9 \pm 0.1$ | $79.6 \pm 0.2$ | $67.2 \pm 0.1$ | $89.4 \pm 0.2$ |
| SimCLR + Ours | 1000 | $50.8 \pm 0.2$ | $77.8 \pm 0.2$ | $67.3 \pm 0.1$ | $89.9 \pm 0.2$ |
| MoCo + Ours | 1000 | $53.9 \pm 0.2$ | $78.9 \pm 0.2$ | $71.2 \pm 0.1$ | $89.5 \pm 0.1$ |
| BYOL + Ours | 1000 | $\mathbf{58.9} \pm 0.2$ | $\mathbf{81.9} \pm 0.2$ | $\mathbf{72.1} \pm 0.2$ | $91.2 \pm 0.1$ |
| Barlow Twins + Ours | 1000 | $57.6 \pm 0.2$ | $80.6 \pm 0.1$ | $68.9 \pm 0.2$ | $\mathbf{91.8} \pm \mathbf{0.2}$ |

epochs, the improvement brought by the proposed method reaches more than 3% on Top-1 and 2% on Top-5 results. This further demonstrates the effectiveness of our method.

**Results on transfer learning tasks.** Table 3 shows the results on the most commonly used object detection and instance segmentation protocol (Chen et al., 2020; Zbontar et al., 2021) for transfer learning, where introducing our method achieves stable improvements in all tasks.

**Results on few-shot learning tasks.** Table 4 shows the effect of the proposed sampling strategy on standard few-shot transfer learning tasks. Compared to the original baselines, introducing our proposed method achieves remarkable performance improvement, achieving more than 5% improvement. These results demonstrate the superiority of the proposed method under data-scarce conditions.

In summary, we can observe that when the SSL methods are trained based on mini-batches generated by our proposed sampling strategy, they all further improve their performance and by at least 2%. This shows that our sampling strategy is effective in reducing the spurious correlations in the distribution of the mini-batch task, which leads to better causal learning and improves the OOD generalization.

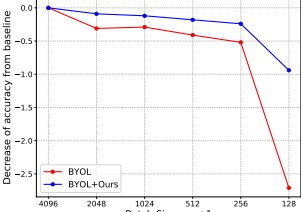

Figure 4: Influence of the hyperparameter $a$.

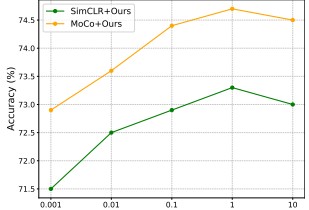

Figure 5: Influence of the hyperparameter $\alpha$.

## 5.3. Ablation Study

**Influence of the batch size hyperparameter a.** As shown in **Theorem** 4.7, a suitable $a$ is important. To explore whether the SSL model is more sensitive to original batch size or $a$, we conduct experiments using ImageNet and BYOL. Figure 4 shows that the performance of BYOL rapidly deteriorates with batch size, while BYOL+Ours remains stable over a wide range of batch sizes. Thus, although the proposed sampling strategy has a high requirement on $a$, the SSL method is less sensitive to $a$.

**Influence of $\alpha$.** In Equation 4, $\alpha$ as a hyperparameter, controls the weight of the term that constrains the orthogonality of the column vectors in the matrix A. It prevents the model from learning redundant or interdependent features, enhancing its generalization. To evaluate its impact, we assess the model performance with varying $\alpha$ (ranging in $[0.001, 0.01, 0.1, 1, 10]$) on ImageNet-100. Figure 5 show that performance peaks at $\alpha = 1$, which is also our setting.

## 6. Related work

**SSL** is an effective unsupervised representation learning paradigm, aimed at learning general representations suitable for various downstream tasks. From (Jaiswal et al., 2020; Kang et al., 2023), existing SSL models can be divided into D-SSL and G-SSL. The D-SSL methods, e.g., SimCLR (Chen et al., 2020), BYOL (Grill et al., 2020a), and Mocov3 (Chen et al., 2021b), are modeled based on the augmentation invariance principle. The G-SSL methods, e.g., MAE (He et al., 2022), iBOT (Zhou et al.), SMA (Xie et al., 2024), are modeled based on the mask and reconstruction principle. In real-world scenarios, the data distribution can shift over time. Thus, improving the OOD generalization of SSL is crucial. While various methods (Ni et al., 2021; Bai et al., 2023; Liu et al., 2022; Qiang et al., 2024; Guo et al., 2024; Li et al., 2022; Song et al., 2024) have been proposed with impressive performance, a remaining challenge is they have to contend with trade-offs between inductive biases or approaches without theoretical guarantees. In this paper, we extend the understanding of SSL from the perspective of task learning and analyze its OOD generalization through causal inference and batch construction.

**Causality Analysis in SSL** plays a crucial role by helping

Table 3: Transfer learning on object detection and instance segmentation with C4-backbone. "AP" is the average precision, "$AP_N$" represents the average precision when the IoU (Intersection and Union Ratio) threshold is $N\%$.

| Method | VOC 07 detection | | | VOC 07+12 detection | | | COCO detection | | | COCO instance segmentation | | |
|---|---|---|---|---|---|---|---|---|---|---|---|---|
| | $AP_{50}$ | AP | $AP_{75}$ | $AP_{50}$ | AP | $AP_{75}$ | $AP_{50}$ | AP | $AP_{75}$ | $AP_{50}^{mask}$ | $AP^{mask}$ | $AP_{75}^{mask}$ |
| Supervised | 74.4 | 42.4 | 42.7 | 81.3 | 53.5 | 58.8 | 58.2 | 38.2 | 41.2 | 54.7 | 33.3 | 35.2 |
| SimCLR (Chen et al., 2020) | 75.9 | 46.8 | 50.1 | 81.8 | 55.5 | 61.4 | 57.7 | 37.9 | 40.9 | 54.6 | 33.3 | 35.3 |
| MoCo (He et al., 2020) | 77.1 | 46.8 | 52.5 | 82.5 | 57.4 | 64.0 | 58.9 | 39.3 | 42.5 | 55.8 | 34.4 | 36.5 |
| BYOL (Grill et al., 2020b) | 77.1 | 47.0 | 49.9 | 81.4 | 55.3 | 61.1 | 57.8 | 37.9 | 40.9 | 54.3 | 33.2 | 35.0 |
| SimSiam (Chen & He, 2021) | 77.3 | 48.5 | 52.5 | 82.4 | 57.0 | 63.7 | 59.3 | 39.2 | 42.1 | 56.0 | 34.4 | 36.7 |
| SwAV (Caron et al., 2020) | 75.5 | 46.5 | 49.6 | 82.6 | 56.1 | 62.7 | 58.6 | 38.4 | 41.3 | 55.2 | 33.8 | 35.9 |
| VICRegL (Bardes et al., 2022) | 75.9 | 47.4 | 52.3 | 82.6 | 56.4 | 62.9 | 59.2 | 39.8 | 42.1 | 56.5 | 35.1 | 36.8 |
| SimCLR + Ours | 77.6 | 50.1 | 51.7 | 85.3 | 58.4 | 63.9 | 59.2 | 40.6 | 43.9 | 57.1 | 35.9 | 37.1 |
| MoCo + Ours | 79.4 | 50.2 | **54.9** | **86.1** | **60.2** | **66.1** | 614 | 42.1 | 44.9 | **59.2** | 36.9 | 38.8 |
| BYOL + Ours | 79.1 | 50.4 | 51.9 | 83.9 | 58.7 | 64.1 | 60.6 | 39.9 | 43.7 | 56.2 | 35.1 | 38.6 |
| SimSiam + Ours | **80.5** | **50.8** | 54.4 | 85.2 | 59.5 | 66.1 | 62.3 | 42.5 | 43.9 | 58.1 | 37.2 | 39.8 |
| SwAV + Ours | 77.9 | 49.3 | 51.8 | 84.9 | 58.1 | 65.8 | 62.1 | 40.2 | 43.9 | 56.9 | 37.3 | 37.9 |
| VICRegL + Ours | 77.9 | 50.4 | 53.9 | 85.2 | 58.8 | 65.3 | **63.1** | **42.2** | **45.3** | 59.1 | **37.8** | **39.9** |

Table 4: Few-shot transfer learning accuracies ($\pm$ 95% confidence interval) on miniImageNet, Omniglot, and CIFAR-FS.

| Method | Omniglot | | | miniImageNet | | | CIFAR-FS | | |
|---|---|---|---|---|---|---|---|---|---|
| | (5,1) | (5,5) | (20,1) | (5,1) | (5,5) | (20,1) | (5,1) | (5,5) | (20,1) |
| SimCLR (Chen et al., 2020) | 90.83 ± 0.21 | 97.67 ± 0.21 | 81.67 ± 0.23 | 42.32 ± 0.38 | 51.10 ± 0.37 | 36.36 ± 0.36 | 49.44 ± 0.30 | 60.02 ± 0.29 | 39.29 ± 0.30 |
| MoCo (He et al., 2020) | 87.83 ± 0.20 | 95.52 ± 0.19 | 80.03 ± 0.21 | 40.56 ± 0.34 | 49.41 ± 0.37 | 36.52 ± 0.38 | 45.35 ± 0.31 | 58.11 ± 0.32 | 37.89 ± 0.32 |
| SwAV (Caron et al., 2020) | 91.28 ± 0.19 | 97.21 ± 0.20 | 82.02 ± 0.20 | 44.39 ± 0.36 | 54.91 ± 0.36 | 37.13 ± 0.37 | 49.39 ± 0.29 | 62.20 ± 0.31 | 40.19 ± 0.32 |
| SimCLR + Ours | 95.05 ± 0.22 | **98.96 ± 0.16** | 91.15 ± 0.20 | 47.14 ± 0.21 | 62.88 ± 0.21 | 39.97 ± 0.16 | **53.18 ± 0.24** | 67.91 ± 0.14 | 46.94 ± 0.21 |
| MoCo + Ours | 93.22 ± 0.19 | 97.93 ± 0.19 | 88.93 ± 0.22 | 46.93 ± 0.21 | 61.22 ± 0.21 | 41.12 ± 0.24 | 51.76 ± 0.22 | 66.42 ± 0.21 | 44.93 ± 0.23 |
| SwAV + Ours | **96.24 ± 0.26** | 98.76 ± 0.22 | **91.96 ± 0.21** | **49.15 ± 0.21** | **64.28 ± 0.29** | **42.22 ± 0.21** | 52.64 ± 0.24 | **70.18 ± 0.21** | **48.19 ± 0.14** |

to identify and understand the underlying relationships between variables. Recent works (Sontakke et al., 2021; Zuo et al., 2021; Qiang et al., 2022; Wang et al., 2024a; Qiang et al., 2023) have focused on developing methods that leverage causal inference to extract more robust feature representations. For instance, (Song et al., 2023) used causal invariance to obtain causal SSL representations; (Von Kügelgen et al., 2021) studied the identifiability of latent representations, aiming to improve efficiency. However, most of them build causal analysis on in-distribution, but ignore the influence of spurious correlations under OOD generalization settings. In this paper, we explore the essential reasons for spurious correlations in SSL and propose a method that makes the relationships between variables free from spurious correlations. Meanwhile, duo to the limited space in the following, we have discussed the content of **Spurious Correlation** in **Appendix** D.

## 7. Conclusion

In this paper, we first establish the connection between mini-batches formed during the SSL training phase and multi-class tasks. Next, we explain the rationale for OOD generalization of SSL from a multi-task learning perspective. We then analyze how existing SSL models, when learning mini-batch tasks, rely on spurious correlations to measure sample similarity, leading to suboptimal performance. This reliance affects the SSL model's approximation of the task distribution, resulting in reduced OOD generalization. We

provide a causal analysis of this issue and theoretically examine the intrinsic reasons for incorporating spurious correlations during the learning process. Based on our causal analysis, we demonstrate that when mini-batches satisfy a specific distribution, e.g., PID, SSL models achieve optimal worst-case OOD performance. This insight guides us to propose a new sampling strategy that ensures the resulting mini-batches satisfy the PID constraints. Extensive theoretical and empirical analyses demonstrate its effectiveness.

## Impact Statement

This paper presents work whose goal is to advance the field of Machine Learning and Self-Supervised Learning. There are many potential societal consequences of our work, none of which we feel must be specifically highlighted here.

## Acknowledgements

The authors would like to thank the anonymous reviewers for their valuable comments. This work is supported in part by the China Postdoctoral Science Foundation, Grant No.2024M753356, and in part by the National Natural Science Foundation of China, No. 62406313.

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

# Appendix

The **Appendices** provide additional details that support the main findings and methods proposed in this paper. It is organized into the following sections:

- **Appendix** A contains the proofs of the presented theorems.

- **Appendix** B provides details for the experimental settings for each experiment.

- **Appendix** C showcases additional experiments that were omitted in the main text due to page limitations.

- **Appendix** D provides the related works for spurious correlation in SSL.

- **Appendix** E explains the differences and connections between task distribution and data distribution.

- **Appendix** F provides the intuitive explanation of several concepts, assumption, and theorems mentioned in the proposed methodology.

- **Appendix** G provides explanation of the identifiability of spurious variable.

## A. Proofs

This section provides the complete proof of Proposition and Theorem in the main text.

### A.1. Proof of Proposition 3.1

Before giving the detailed proofs of Proposition 3.1, we first provide the problem definition. Given multiple pairs of samples in an SSL task, let $x^{\text{label}}$ be the anchor of a specific pair, then the remaining samples involving two classes of being $x^{\text{label}}$ and not $x^{\text{label}}$. Let $x^{\text{label}}$ and $\bar{x}^{\text{label}}$ represent the label variables of being $x^{\text{label}}$ and not $x^{\text{label}}$, since these are binary classification tasks, $x^{\text{label}}$ and $\bar{x}^{\text{label}}$ belong to the set $\pm 1$. Note that any multi-classification task can be decomposed into binary tasks.

We assume that the labels are drawn from two different probabilities, with balanced sampling probabilities for label values, i.e., $P(x^{\text{label}} = 1) = P(x^{\text{label}} = -1) = 0.5$. Our conclusions also hold for imbalanced distributions. Next, we consider two $d$-dimensional factors $F_{x+}$ and $F_s$ representing the knowledge to tackle the two labels. Both are drawn from the Gaussian distribution:

$$F_{x+} \sim \mathcal{N}(x^{\text{label}} \cdot \mu_{\text{label}}, \sigma_{\text{label}}^2 I)$$

$$F_s \sim \mathcal{N}(\bar{x}^{\text{label}} \cdot \mu_s, \sigma_s^2 I)$$

where $\mu_{\text{label}}, \mu_s \in \mathbb{R}^{N_s}$ denote the mean vectors, while $\sigma_{\text{label}}^2$ and $\sigma_s^2$ denote the covariance vectors. We examine the spurious correlations in SSL. To simplify our analysis, we define $p_{sc}$ as the varying correlations that result from different spurious correlations across batches.

Training a single model will result in the optimal model for the target, incorporating non-causal features from the other sample pairs. To substantiate this, we derive the optimal SSL model as follows:

$$P(x^{\text{label}}|F_{x+}, F_s) = \frac{P(x^{\text{label}}, F_{x+}, F_s)}{P(F_{x+}, F_s)} = \frac{P(x^{\text{label}}, F_{x+}, F_s)}{\sum_{x^{\text{label}} \in \{-1,1\}} P(x^{\text{label}}, F_{x+}, F_s)}.$$

Using the independence of $F_s$ and $F_{x+}$ given $x^{\text{label}}$, we rewrite the joint probability $P(x^{\text{label}}, F_{x+}, F_s)$ as:

$$
\begin{aligned}
P(x^{\text{label}}, F_{x+}, F_s) &= P(x^{\text{label}}, F_{x+}) \cdot P(F_s|x^{\text{label}}, F_{x+}) \\
&= P(x^{\text{label}}, F_{x+}) \cdot P(F_s|x^{\text{label}}) \\
&= P(x^{\text{label}}, F_{x+}) \cdot \sum_{\bar{x}^{\text{label}} \in \{-1,1\}} P(F_s, \bar{x}^{\text{label}}|x^{\text{label}}) \\
&= P(x^{\text{label}})P(F_{x+}|x^{\text{label}}) \cdot \sum_{\bar{x}^{\text{label}} \in \{-1,1\}} P(F_s|\bar{x}^{\text{label}})P(\bar{x}^{\text{label}}|x^{\text{label}})
\end{aligned}
$$

Assuming that $F_{x^+}$ and $F_s$ are drawn from Gaussian distributions, and $P(Y_{i/j}, F_{x^+}, F_s) = \text{sigmoid}\left(\frac{\mu_{\text{label}}}{\sigma_{\text{label}}^2} F_{x^+} + \frac{\mu_s}{\sigma_s^2} F_s\right)$, where $\frac{\mu_{\text{label}}}{\sigma_{\text{label}}^2}$ and $\frac{\mu_s}{\sigma_s^2}$ are the regression vectors for the optimal Bayesian classifier, we have:

$$
\begin{aligned}
P(x^{\text{label}}, F_{x^+}, F_s) &= P(x^{\text{label}}, F_{x^+}) \cdot P(F_s | x^{\text{label}}, F_{x^+}) \\
&= P(x^{\text{label}}) P(F_{x^+} | x^{\text{label}}) \cdot \sum_{\bar{x}^{\text{label}} \in \{-1, 1\}} P(F_s | \bar{x}^{\text{label}}) P(\bar{x}^{\text{label}} | x^{\text{label}}) \\
&\propto e^{x^{\text{label}} \cdot \frac{\mu_{\text{label}}}{\sigma_{\text{label}}^2} F_{x^+}} \left( p_{sc} e^{x^{\text{label}} \cdot \frac{\mu_s}{\sigma_s^2} F_s} + (1 - p_{sc}) e^{-x^{\text{label}} \cdot \frac{\mu_s}{\sigma_s^2} F_s} \right) \\
&= p_{sc} e^{x^{\text{label}} \cdot \left( \frac{\mu_{\text{label}}}{\sigma_{\text{label}}^2} F_{x^+} + \frac{\mu_s}{\sigma_s^2} F_s \right)} + (1 - p_{sc}) e^{x^{\text{label}} \cdot \left( \frac{\mu_{\text{label}}}{\sigma_{\text{label}}^2} F_{x^+} - \frac{\mu_s}{\sigma_s^2} F_s \right)}
\end{aligned}
$$

Let:

$$
\begin{aligned}
\beta^+ &= \frac{\mu_{\text{label}}}{\sigma_{\text{label}}^2} F_{x^+} + \frac{\mu_s}{\sigma_s^2} F_s \\
\beta^- &= \frac{\mu_{\text{label}}}{\sigma_{\text{label}}^2} F_{x^+} - \frac{\mu_s}{\sigma_s^2} F_s
\end{aligned}
$$

Substituting $\beta^+$ and $\beta^-$ back into the original equation, we have:

$$
P(x^{\text{label}} | F_{x^+}, F_s) = \frac{1}{1 + \frac{p_{sc} e^{x^{\text{label}} \cdot \beta^+} + (1 - p_{sc}) e^{x^{\text{label}} \cdot \beta^-}}{p_{sc} e^{-x^{\text{label}} \cdot \beta^+} + (1 - p_{sc}) e^{-x^{\text{label}} \cdot \beta^-}}}
$$

When the samples are easy to distinguish, e.g., the similarity of the augmented sample from different pairs is not 1:

$$
P(x^{\text{label}} | F_{x^+}, F_s) = \frac{1}{1 + e^{x^{\text{label}} \cdot (\beta^+ + \beta^-)}}
$$

Combining with the expressions for $\beta^+$ and $\beta^-$, we get:

$$
P(x^{\text{label}} | F_{x^+}, F_s) = \frac{1}{1 + e^{2x^{\text{label}} \cdot \left( \frac{\mu_{\text{label}}}{\sigma_{\text{label}}^2} F_{x^+} \right)}}
$$

In this case, the optimal SSL model only utilizes its own factor $F_{x^+}$ and assigns zero weight to the non-causal factor $F_s$ from task $\tau_j$. Thus, if it is difficult to distinguish between the different pairs, the optimal model has non-zero weights for non-causal factors for each task. When the samples are difficult to distinguish, e.g., in the most extreme case, the similarity of the augmented sample from different pairs is equal to 1, we have:

$$
P(x^{\text{label}} | F_{x^+}, F_s) = \frac{1}{1 + e^{2x^{\text{label}} \cdot \beta^+}}
$$

Combining with the expressions for $\beta^+$ and $\beta^-$, we get:

$$
P(x^{\text{label}} | F_{x^+}, F_s) = \frac{1}{1 + e^{2x^{\text{label}} \cdot \left( \frac{\mu_{\text{label}}}{\sigma_{\text{label}}^2} F_{x^+} + \frac{\mu_s}{\sigma_s^2} F_s \right)}}
$$

In this case, the optimal classifier incorporates both factors $F_{x^+}$ and $F_s$. Thus, if $p_{sc} \neq 0.5$, the optimal classifier assigns non-zero weights to non-causal factors for each task.

### A.2. Proof of Theorem 3.4

*Proofs.* Here, we provide proof of the minimax optimality of the SSL model trained on PID. The SSL model trained on PID $p^{\text{PI}}(x^+, x^{\text{label}})$ has $p_f(x^{\text{label}} | x^+) = p^{\text{PI}}(x^{\text{label}} | x^+)$. Now, consider the expected cross-entropy loss of this classifier on an

unseen test distribution $p^e$:

$$\mathcal{L}^e(p^{\mathrm{PI}}(x^{\mathrm{label}}|x^+)) = -\mathbb{E}_{p^e(x^+,x^{\mathrm{label}})} \log p^{\mathrm{PI}}(x^{\mathrm{label}}|x^+)$$

$$= -\mathbb{E}_{p^e(x^+,x^{\mathrm{label}})} \log p^{\mathrm{PI}}(x^{\mathrm{label}}) + \mathbb{E}_{p^e(x^+,x^{\mathrm{label}})} \log \frac{p^{\mathrm{PI}}(x^{\mathrm{label}})}{p^{\mathrm{PI}}(x^{\mathrm{label}}|x^+)}$$

$$= \mathcal{L}^e(p^{\mathrm{PI}}(x^{\mathrm{label}})) + \mathbb{E}_{p^e(X,x^{\mathrm{label}},s)} \left[ \log \frac{p^{\mathrm{PI}}(x^{\mathrm{label}})}{p^{\mathrm{PI}}(x^{\mathrm{label}}|x^+)} \right]$$

$$= \mathcal{L}^e(p^{\mathrm{PI}}(x^{\mathrm{label}})) + \mathbb{E}_{p^e(x^{\mathrm{label}},s)} \left[ \mathbb{E}_{p^{\mathrm{PI}}(X|x^{\mathrm{label}},s)} \left[ \log \frac{p^{\mathrm{PI}}(x^{\mathrm{label}})}{p^{\mathrm{PI}}(x^{\mathrm{label}}|x^+)} \right] \right]$$

Consider that $x^{\mathrm{label}} \perp\!\!\!\perp_{\mathrm{PI}} s$ and $x^{\mathrm{label}} \perp\!\!\!\perp_{\mathrm{PI}} s|x^+$, we get:

$$\mathcal{L}^e(p^{\mathrm{PI}}(x^{\mathrm{label}}|x^+)) = \mathcal{L}^e(p^{\mathrm{PI}}(x^{\mathrm{label}})) + \mathbb{E}_{p^e(x^{\mathrm{label}},s)} \left[ \mathbb{E}_{p^{\mathrm{PI}}(x^+|x^{\mathrm{label}},s)} \left[ \log \frac{p^{\mathrm{PI}}(x^{\mathrm{label}}|s)}{p^{\mathrm{PI}}(x^{\mathrm{label}}|x^+,s)} \right] \right]$$

$$= \mathcal{L}^e(p^{\mathrm{PI}}(x^{\mathrm{label}})) + \mathbb{E}_{p^e(x^{\mathrm{label}},s)} \left[ \mathbb{E}_{p^{\mathrm{PI}}(x^+|x^{\mathrm{label}},s)} \left[ \log \frac{p^{\mathrm{PI}}(x^+|s)}{p^{\mathrm{PI}}(x^+|x^{\mathrm{label}},s)} \right] \right]$$

$$= \mathcal{L}^e(p^{\mathrm{PI}}(x^{\mathrm{label}})) - \mathbb{E}_{p^e(x^{\mathrm{label}},s)} \mathrm{KL}[p^{\mathrm{PI}}(x^+|x^{\mathrm{label}},s)||p^{\mathrm{PI}}(x^+|s)].$$

Thus we have the cross entropy loss of $p^{\mathrm{PI}}(x^+, x^{\mathrm{label}})$ in any environment $e$ is smaller than that of $p^{\mathrm{PI}}(x^{\mathrm{label}}) = \frac{1}{m}$ (random guess):

$$\mathcal{L}^e(p^{\mathrm{PI}}(x^{\mathrm{label}}|x^+)) - \mathcal{L}^e(p^{\mathrm{PI}}(x^{\mathrm{label}})) \leq -\mathbb{E}_{p^e(x^{\mathrm{label}},s)} \mathrm{KL}[p^{\mathrm{PI}}(x^+|x^{\mathrm{label}},s)||p^{\mathrm{PI}}(x^+|s)] \leq 0,$$

which means:

$$\max_{e' \in \mathcal{E}} \left[ \mathcal{L}^{e'}(p^{\mathrm{PI}}(x^{\mathrm{label}}|x^+)) - \mathcal{L}^{e'}(p^{\mathrm{PI}}(x^{\mathrm{label}})) \right] \leq 0.$$

where the performance of $p^{\mathrm{PI}}(x^+, x^{\mathrm{label}})$ is at least as good as a random guess in any environment. Since we assume the environment diversity, that is for any $p^e$ with $x^{\mathrm{label}} \perp\!\!\!\perp_e s$, there exists an environment $e'$ such that $p^e(x^{\mathrm{label}}|x^+)$ performs worse than a random guess. So we have:

$$\max_{e' \in \mathcal{E}} \left[ \mathcal{L}^{e'}(p^{\mathrm{PI}}(x^{\mathrm{label}}|x^+)) - \mathcal{L}^{e'}(p^{\mathrm{PI}}(x^{\mathrm{label}})) \right] \leq 0 < \max_{e' \in \mathcal{E}} \left[ \mathcal{L}^{e'}(p^e(x^{\mathrm{label}}|x^+)) - \mathcal{L}^{e'}(p^{\mathrm{PI}}(x^{\mathrm{label}})) \right].$$

This inequality shows that the model trained on PID does not perform worse than the random guess model. The KL divergence term is non-negative, ensuring that the difference between the loss of the PID-trained model and the random guess model is non-positive. Now we want to prove that we can obtain $p^e(x^{\mathrm{label}}|x^+) = p^{\mathrm{PI}}(x^{\mathrm{label}}|x^+)$ for any $s \in \mathcal{S}$ with $\forall e \in \mathcal{E}$, $x^{\mathrm{label}} \perp\!\!\!\perp_e s$, $x^{\mathrm{label}} \perp\!\!\!\perp_e s|x^+$, $p^e(x^{\mathrm{label}}) = \frac{1}{m}$, we have:

$$p^e(x^{\mathrm{label}}|x^+) = p^e(x^{\mathrm{label}}|x^+,s) = p^e(x^{\mathrm{label}}) \frac{p^e(x^+|x^{\mathrm{label}},s)}{\mathbb{E}_{p^e(x^{\mathrm{label}}|s)}[p^e(x^+|s,x^{\mathrm{label}})]}$$

$$= p^{\mathrm{PI}}(x^{\mathrm{label}}) \frac{p^{\mathrm{PI}}(x^+|x^{\mathrm{label}},s)}{\mathbb{E}_{p^{\mathrm{PI}}(x^{\mathrm{label}})}[p^{\mathrm{PI}}(x^+|s,x^{\mathrm{label}})]} = p^{\mathrm{PI}}(x^{\mathrm{label}}|x^+,s) = p^{\mathrm{PI}}(x^{\mathrm{label}}|x^+).$$

Thus, we have the following minimax optimality $p^{\mathrm{PI}}(x^{\mathrm{label}}|x^+) = \arg\min_{p_f \in \mathcal{F}} \max_{e \in \mathcal{E}} \mathcal{L}^e(p_\psi(x^{\mathrm{label}}|x^+))$. Based on the above analyses, we have $f^*$ is the minimax optimal across all elements in $\mathcal{D}$, e.g., $f^* = \arg_f \min \max_{e \in \mathcal{D}} \mathcal{L}^e(p_f(x^{\mathrm{label}}|x^+))$.

### A.3. Proof of Theorem 4.3

*Proofs.* We aim to establish the identifiability of the key parameters in the VAE that govern the spuriously correlated covariate features. The proof proceeds through the following two parts: (i) Introducing Auxiliary Variables: We introduce both $e$ and $x^{\mathrm{label}}$ as auxiliary variables to better capture the dependency structure. (ii) Incorporating Causal Mechanism: We specify the causal mechanism that generates $x^+$ as $x = \mathrm{f}(x^{\mathrm{label}}, s) + \epsilon = \mathrm{f}_x^{\mathrm{label}}(x) + \epsilon$, where $\epsilon$ represents the noise term.

First, we transform the equality of the marginal distributions over the observed data into the equality of a noise-free distribution. Let us start by considering two sets of model parameters, i.e., $\theta = (f, g, A)$ and $\theta' = (f', g', A')$. Suppose that the marginal distributions of the observed data are equal across these two parameterizations:

$$p_\theta(x^+|x^{\text{label}}, e) = p_{\theta'}(x^+|x^{\text{label}}, e) \quad \forall e \in \mathcal{E}_{\text{train}}.$$

This condition implies that for each $e \in \mathcal{E}_{\text{train}}$, the distributions of $x^+$ conditioned on $x^{\text{label}}$ and $e$ must be the same for both sets of parameters. This leads us to the following equation:

$$\int_{\mathcal{Z}} p_{g,A}(Z|x^{\text{label}}, e)p_f(x^+|Z, x^{\text{label}})dZ = \int_{\mathcal{Z}} P_{g',A'}(Z|x^{\text{label}}, e)p'_f(x^+|Z, x^{\text{label}})dZ$$

Now, the noise term $p_\epsilon(x^+ - f_x^{\text{label}}(Z))$ appears in both expressions, as it governs the relationship between $x^+$ and the latent variable $Z$. This results in:

$$\int_{\mathcal{Z}} p_{g,A}(Z|x^{\text{label}}, e)p_\epsilon(x^+ - f_x^{\text{label}}(Z))dZ = \int_{\mathcal{Z}} p_{g',A'}(Z|x^{\text{label}}, e)p_\epsilon(x^+ - f_x^{'\text{label}}(Z))dZ$$

Next, we define the volume of the matrix A as $\text{vol}(A) := \sqrt{\det(A^\top A)}$ and introduce a change of variables in the integrals. On the left-hand side, we change variables to $x^+ = f_x^{\text{label}}(Z)$, while on the right-hand side, we set $\bar{x}^+ = \bar{f}_x^{\text{label}}(Z)$. Since $f$ is injective, we can write $f^{-1}(\bar{x}^+) = (x^{\text{label}}, Z)$, which implies that $Z$ can be recovered from $x^+$ and $x^{\text{label}}$. Thus, we arrive at the following equality:

$$\int_{\mathbb{R}^d} \tilde{p}_{g,A,f,x^{\text{label}},e}(\bar{x}^+)p_\epsilon(x^+ - \bar{x}^+)\, d\bar{x}^+ = \int_{\mathbb{R}^d} \tilde{p}_{g',A',f',x^{\text{label}},e}(\bar{x}^+)p_\epsilon(x^+ - \bar{x}^+)\, d\bar{x}^+.$$

The key insight here is that both sides describe the same transformation, albeit with different parameterizations.

We now introduce a convolution representation for the probability distributions. Specifically, define:

$$\tilde{p}_{g,A,f,x^{\text{label}},e}(x^+) = p_{g,A}(f_x^{\text{label}^{-1}}(x^+)|x^{\text{label}}, e)\text{vol}J_{f_x^{\text{label}-1}}(x^+)1_{\S^+}(x^+),$$

where $J_{f_x^{\text{label}-1}}(x^+)$ denotes the Jacobian of $f_x^{\text{label}^{-1}}$. Similarly, we define the corresponding expression on the right-hand side. We then introduce the convolution operator $*$ and apply the Fourier transform to both sides:

$$(\tilde{p}_{g,A,f,x^{\text{label}},e} * p_\epsilon)(x^+) = (\tilde{p}_{g',A',f',x^{\text{label}},e} * p_\epsilon)(x^+).$$

Then, we use $*$ for the convolution operator, and use $F[\cdot]$ to designate the Fourier transform. The characteristic function of $\epsilon$ is then $\phi_\epsilon = F[p_\epsilon]$. Exploit the properties of the Fourier transform to transform the convolution into a multiplication. This means that in the Fourier domain, we have $F[(\tilde{p}_{g,A,f,x^{\text{label}},e} * p_\epsilon)(x^+)] = F[\tilde{p}_{g,A,f,x^{\text{label}},e}](\omega) \cdot F[p_\epsilon](\omega)$ Meanwhile, we dropped $\phi_\epsilon(\omega)$ from both sides as it is non-zero almost everywhere (by assumption of the Theorem).

$$\tilde{p}_{g,A,f,x^{\text{label}},e}(x^+) = \tilde{p}_{g',A',f',x^{\text{label}},e}(x^+),$$

which shows that the probability distributions for $x^+$ under the two parameterizations are equal.

Obtaining the above results, we simplify the expression by removing terms that depend on $x^+$, $x^{\text{label}}$, or $e$, and then take the logarithm of both sides of the equation. This results in:

$$\log \text{vol}J_{f^{-1}}(x^+) + \sum_{i=1}^{n}(\log Q_i(f_i^{-1}(x^+)) - \log W_i^e(x^{\text{label}}) + \sum_{j=1}^{k}T_{i,j}(f_i^{-1}(x^+))\lambda_{i,j}^e(x^{\text{label}}))$$

$$= \log \text{vol}J_{f'^{-1}}(x^+) + \sum_{i=1}^{n}(\log Q'_i(f_i'^{-1}(x^+)) - \log W_i'^e(x^{\text{label}}) + \sum_{j=1}^{k}T'_{i,j}(f_i'^{-1}(x^+))\lambda_{i,j}'^e(x^{\text{label}})).$$

Next, we evaluate this equation at several points $(e_0, x_0^{\text{label}}), (e_1, x_1^{\text{label}}), ..., (e_{nk}, x_{nk}^{\text{label}})$ provided by assumption (3) of the theorem. We evaluate the above equations at these points to obtain $k + 1$ equations, and subtract the first equation from the

remaining $k$ equations to obtain:

$$\langle T(f^{-1}(x^+)), \lambda^{e_l}(x_l^{\text{label}}) - \lambda^{e_0}(x_0^{\text{label}})\rangle + \sum_{i=1}^{n} \log \frac{W_i^{e_0}(x_0^{\text{label}})}{W_i^{e_l}(x_l^{\text{label}})}$$

$$= \langle T'(f^{-1}(x^+)), \lambda'^{e_l}(x_l^{\text{label}}) - \lambda'^{e_0}(x_0^{\text{label}})\rangle + \sum_{i=1}^{n} \log \frac{W_i'^{e_0}(x_0^{\text{label}})}{W_i'^{e_l}(x_l^{\text{label}})}.$$

This creates a system of equations, which will be solved in the next step.

Let $\mathcal{L}$ be the matrix defined in assumption (3) and $\mathcal{L}'$ similarly defined for $\lambda'$ ($\mathcal{L}'$ is not necessarily invertible). Define $b_l = \sum_{i=1}^{n} \log \frac{W_i'^{e_0}(x_0^{\text{label}})W_i^{e_l}(x_l^{\text{label}})}{W_i^{e_0}(x_0^{\text{label}})W_i'^{e_l}(x_l^{\text{label}})}$ and $b = [b_l]_{l=1}^{nk}$.

Then, we can obtain the matrix form:

$$\mathcal{L}^T T(f^{-1}(x^+)) = \mathcal{L}'^T T'(f'^{-1}(x^+)) + b.$$

We multiply both sides of Equation 6 by $\mathcal{L}^{-T}$ to get:

$$T(f^{-1}(x^+)) = AT'(f'^{-1}(x^+)) + c.$$

Where $A = \mathcal{L}^{-T}\mathcal{L}'$ and $c = \mathcal{L}^{-T}b$. To complete the proof, we must demonstrate that A is invertible. By the definition of $T$, its Jacobian exists and is an $nk \times n$ matrix with rank $n$. Consequently, the Jacobian of $T' \circ f'^{-1}$ also exists and has rank $n$, which implies that $A$ is of rank $n$ as well. We mainly consider two cases: (i) If $k = 1$, then A is invertible since $A \in \mathbb{R}^{n \times n}$; and (ii) If $k > 1$, define $\bar{x} = f^{-1}(\mathbf{x})$ and $T_i(\bar{x}_i) = (T_{i,1}(\bar{x}_i), \ldots, T_{i,k}(\bar{x}_i))$.

Suppose for any choice of $\bar{x}_i^1, \bar{x}_i^2, \ldots, \bar{x}_i^k$, the family $\left(\frac{dT_i(\bar{x}_i^1)}{d\bar{x}_i^1}, \ldots, \frac{dT_i(\bar{x}_i^k)}{d\bar{x}_i^k}\right)$ is never linearly independent. This implies that $T_i(\mathbb{R})$ lies within a subspace of $\mathbb{R}^k$ with a dimension of at most $k - 1$. Let $h$ be a non-zero vector orthogonal to $T_i(\mathbb{R})$. Then for all $x \in \mathbb{R}$, we have $\left\langle \frac{dT_i(x)}{dx}, h\right\rangle = 0$. By integrating, we find that $\langle T_i(x), h\rangle = \text{const}$. Since this holds for all $x \in \mathbb{R}$ and $h \neq 0$, we conclude that the distribution is not strongly exponential. Thus, by contradiction, there must exist $k$ points $\bar{x}_i^1, \bar{x}_i^2, \ldots, \bar{x}_i^k$ such that $\left(\frac{dT_i(\bar{x}_i^1)}{d\bar{x}_i^1}, \ldots, \frac{dT_i(\bar{x}_i^k)}{d\bar{x}_i^k}\right)$ are linearly independent.

Next, collect these points into $k$ vectors $(\bar{x}^1, \ldots, \bar{x}^k)$ and concatenate the $k$ Jacobians $J_T(\bar{x}^l)$ evaluated at each of those vectors horizontally into the matrix $Q = (J_T(\bar{x}^1), \ldots, J_T(\bar{x}^k))$. Similarly, define $Q'$ as the concatenation of the Jacobians of $T'(f'^{-1} \circ f(\bar{x}))$ evaluated at those points. Then the matrix $Q$ is invertible. By differentiating Equation 6 for each $x^l$, we get $Q = AQ'$ The invertibility of $Q$ implies the invertibility of A and $Q'$. This completes the proof.

### A.4. Proof of Theorem 4.7

*Proofs.* In **Algorithm** 1, we describe the procedure for uniformly sampling $a$ different labels. Specifically, we define this process as sampling $x_{\text{alt}}^{\text{label}} = \{x_1^{\text{label}}, x_2^{\text{label}}, \ldots, x_a^{\text{label}}\}$ using the following procedure:

$$x_1^{\text{label}} \sim U\{1, 2, \ldots, mu\} \setminus \{x_e^{\text{label}}\}$$
$$x_2^{\text{label}} \sim U\{1, 2, \ldots, mu\} \setminus \{x_e^{\text{label}}, x_1^{\text{label}}\}$$
$$\vdots$$
$$x_a^{\text{label}} \sim U\{1, 2, \ldots, mu\} \setminus \{x_e^{\text{label}}, x_1^{\text{label}}, x_2^{\text{label}}, \ldots, x_{a-1}^{\text{label}}\},$$

Here, $U$ represents the uniform distribution over the specified set, and $x_e^{\text{label}}$ is the label selected in the current training instance, from which we exclude during the sampling of other labels.

Let $\mathcal{D}_{\text{balanced}}$ denote the balanced data distribution, which is drawn from the distribution $\hat{p}^B(x^+, x^{\text{label}})$. On the other hand, let $\mathcal{D}^e$ be the data distribution corresponding to the environmental distribution $p(x^+, x^{\text{label}})$. We assume that for every balancing score match, the labels in the balanced dataset are perfectly matched. Under this assumption, for all environments $e \in \mathcal{E}_{\text{train}}$, we have:

$$\hat{p}^B(x^{\text{label}}|ba^e(s)) = p(x^{\text{label}}|ba^e(s)),$$

where $ba^e(s)$ denotes the balancing score specific to environment $e$ and sample $s$.

By the definition of a balancing score, $p(x^{\text{label}}|s) = p(x^{\text{label}}|ba^e(s))$ and $\hat{p}^B(x^{\text{label}}|s) = \hat{p}^B(x^{\text{label}}|ba^e(s))$, then we have:

$$\hat{p}^B(x^{\text{label}}|s) = p(x^{\text{label}}|s).$$

and thus, by substitution, we have $\hat{p}^B(x^{\text{label}}|s) = \hat{p}^B(x^{\text{label}}|ba^e(s))$. Consequently, we conclude:

$$\hat{p}^B(x^{\text{label}}|s) = p(x^{\text{label}}|s).$$

This implies that the balanced distribution $\hat{p}^B(x^{\text{label}}|s)$ is identical to the original data distribution $p(x^{\text{label}}|s)$. Therefore, the joint distribution of the balanced data is:

$$\hat{p}^B(x^+, x^{\text{label}}, s) = p^B(x^+, x^{\text{label}}, s).$$

Finally, this shows that the balanced dataset $\mathcal{D}_{\text{balanced}}$ can be regarded as being sampled from a PID (Permuted and Injected Distribution). The implication is that the balanced dataset behaves in the same manner as the original distribution, but with some permutations introduced by the balancing procedure.

## B. Experimental Settings

In this section, we provide the details of the settings and datasets for each experiment.

**Unsupervised Learning** Following the widely adopted protocol (Chen et al., 2020; Wang et al., 2024b), we freeze the feature extractor and train a supervised linear classifier on top of it. The Adam optimizer is used, with Momentum set to 0.8 and weight decay set to $10^{-4}$. The linear classifier is trained for 500 epochs, with a batch size of 128. The learning rate starts at $5 \times 10^{-2}$ and decays to $5 \times 10^{-6}$. For this experiment, we utilize several benchmark datasets to evaluate the model's performance. CIFAR-10 and CIFAR-100 are small-scale image classification datasets consisting of 60,000 32×32 color images in 10 and 100 classes, respectively. STL-10 is another small-scale dataset that contains 100,000 unlabeled images and 5,000 labeled examples from 10 classes, with a higher image resolution (96×96). Tiny ImageNet contains 100,000 64×64 images across 200 classes and serves as a more challenging small-scale benchmark. For these datasets, we use ResNet-18 as the feature extractor. For larger datasets, we employ ImageNet-100 (a subset of ImageNet with 100 classes) and the full ImageNet dataset, which consists of over 1.2 million images in 1,000 classes, using ResNet-50 as the feature extractor.

**Semi-Supervised Learning** In accordance with the standard protocol (Zbontar et al., 2021), we create two balanced subsets by sampling 1% and 10% of the training dataset. Specifically, we use the ImageNet dataset, a large-scale benchmark for visual recognition tasks, comprising 1.2 million images in 1,000 categories. The subsets contain 1% and 10% of the labeled training data, which are used for fine-tuning the model. The models are fine-tuned for 50 epochs, with learning rates set to 0.05 and 1.0 for the classifier and 0.0001 and 0.01 for the backbone on the 1% and 10% subsets, respectively.

**Transfer Learning** We conduct three transfer learning experiments, including object detection and instance segmentation, transfer to other domains, and video-based tasks. For object detection, we evaluate the model on two benchmark datasets: Pascal VOC and COCO. Pascal VOC is widely used for object detection tasks, containing around 20,000 images across 20 categories. We train a Faster R-CNN (Ren et al., 2015) model on the combined VOC 2007 and 2012 datasets (VOC 07+12), which contains around 16,000 images, and adjust the learning rate at 18K and 22K iterations. We also conduct experiments on a smaller version of Pascal VOC, the VOC 07 set (5K images), with a reduced number of iterations. For instance segmentation, we use the COCO 2017 dataset, which contains over 118,000 images and covers 80 object categories. We train a Mask R-CNN (He et al., 2017) with the standard 1× schedule and C4-backbone (Wu et al., 2019), reporting results on the validation split.

**Few-shot Learning** The protocol outlined in (Wang et al., 2024b; 2023b) is followed for few-shot learning, where we evaluate the proposed method on three standard few-shot learning benchmarks: miniImageNet, Omniglot, and CIFAR-FS. miniImageNet is a widely used few-shot learning benchmark derived from the ImageNet dataset, consisting of 60,000 84×84 images across 100 classes. Omniglot is a dataset designed for character recognition, containing 1,623 different characters from 50 different alphabets, making it suitable for testing few-shot learning algorithms. CIFAR-FS is a few-shot version of the CIFAR-100 dataset, specifically adapted for few-shot learning tasks, containing 100 classes with 600 images per class. For each task, $N$ samples without class-level overlap are randomly selected, and $K$-times data augmentation is applied to

create an $N$-way $K$-shot task. The model is optimized using stochastic gradient descent (SGD) with momentum and weight decay values set to 0.9 and $10^{-4}$, respectively. The trained model's performance is then evaluated on unseen samples drawn from new classes, testing its ability to generalize in few-shot scenarios.

## C. Additional Experiments

### C.1. Evaluation on Generative SSL

To examine the model's impact on generating SSL, we conducted a series of experiments using the ImageNet-1K dataset (Deng et al., 2009). We started with self-supervised pre-training on the ImageNet-1K (IN1K) training set. Next, we evaluated the representations through supervised training using two methods: (i) end-to-end fine-tuning and (ii) linear probing. We reported the top-1 validation accuracy for a single 224×224 crop. For these experiments, we employed ViT-Large (ViT-L/16) (Dosovitskiy et al., 2020) as the backbone. ViT-Large is significantly larger (an order of magnitude bigger) than ResNet-50 (He et al., 2016) and has a tendency to overfit. The following section provides a comparison of the models.

Table 5: Comparison between models.

| Method | scratch, original | scratch, our impl. | baseline MAE | MAE + Ours |
|---|---|---|---|---|
| Top 1 | 76.5 | 82.5 | 84.9 | 86.4 |

Table 6: Comparisons with previous results on ImageNet-1K using the ImageNet-1K training set for pre-training, except for the tokenizer in BEiT, which was pre-trained on 250M DALLE data (Ramesh et al., 2021).

| Method | pre-train data | ViT-B | ViT-L | ViT-H | ViT-H$_{448}$ |
|---|---|---|---|---|---|
| DINO | IN1K | 82.8 | - | - | - |
| MoCo | IN1K | 83.2 | 84.1 | - | - |
| BEiT | IN1K+DALLE | 83.2 | 85.2 | - | - |
| MAE | IN1K | 83.6 | 85.9 | 86.9 | 87.8 |
| MAE+Ours | IN1K | 85.9 | 87.4 | 88.6 | 89.3 |

**Comparisons with self-supervised methods.** In **Table** 6 we compare the fine-tuning results of self-supervised ViT models. Our method has shown steady improvement from bigger models. We obtain 88.6% accuracy using ViT-H (224 size). The previous best accuracy, among all methods, using only IN1K data, is 87.1% (512 size) (Yuan et al., 2022), based on advanced networks. We improve over the state-of-the-art by a nontrivial margin in the highly competitive benchmark of IN1K (no external data). Our result is based on vanilla ViT, and we expect advanced networks will perform better.

**Object detection and segmentation.** We fine-tune Mask R-CNN (He et al., 2017) end-to-end on COCO (Lin et al., 2014). The ViT backbone is adapted for use with FPN (Lin et al., 2017). We apply this approach to all entries in Table 3. We report box AP for object detection and mask AP for instance segmentation. Compared to supervised pre-training, our MAE performs better under all configurations (**Table** 7). Meanwhile, we also transfer the learned VideoMAE + Ours on Kinetics-400 (Kay et al., 2017) to the downstream action detection dataset AVA (Murray et al., 2012). Following the standard setting, we evaluate on top 60 common classes with mean Average Precision (mAP) as the metric. The results are

Table 7: COCO object detection and segmentation using a ViT Mask R-CNN baseline.

| Method | pre-train data | AP$^{box}$ | | AP$^{mask}$ | |
|---|---|---|---|---|---|
| | | ViT-B | ViT-L | ViT-B | ViT-L |
| supervised | IN1K w/ labels | 47.9 | 49.3 | 42.9 | 43.9 |
| MoCo v3 | IN1K | 47.9 | 49.3 | 42.7 | 44.0 |
| BEiT | IN1K+DALLE | 49.8 | 53.3 | 44.4 | 47.1 |
| MAE | IN1K | 50.3 | 53.3 | 44.9 | 47.2 |
| MAE + Ours | IN1K | 52.5 | 55.9 | 46.4 | 49.7 |

Table 8: Performance on for text recognition.

| Methods | IIIT5K | IC03 |
|---|---|---|
| SimCLR (Chen et al., 2020) | 1.7 | 3.8 |
| SeqCLR (Aberdam et al., 2021) | 35.7 | 43.6 |
| SimCLR + Ours | 18.7 | 19.0 |
| SeqCLR + Ours | 38.5 | 47.4 |

Table 9: Performance on Kinetics-700. "Extra Labels" denotes that if the pre-trained models are additionally fine-tuned on the pre-training dataset with labels before being transferred to AVA. $T \times \tau$ refers to the frame number and sample rate.

| Method | Backbone | Pre-train Dataset | Extra Labels | $T \times \tau$ | GFLOPs | Param | mAP |
|---|---|---|---|---|---|---|---|
| VideoMAE | ViT-L | Kinetics-700 | No | $16 \times 4$ | 597 | 305 | 36.1 |
| VideoMAE | ViT-L | Kinetics-700 | Yes | $16 \times 4$ | 597 | 305 | 39.3 |
| VideoMAE + Ours | ViT-L | Kinetics-700 | No | $16 \times 4$ | 597 | 305 | 38.7 |
| VideoMAE + Ours | ViT-L | Kinetics-700 | Yes | $16 \times 4$ | 597 | 305 | 42.1 |

shown in **Table 9**. From the results, we can observe that our method achieves stable improvements.

**Evaluation for VAE** To illustrate without loss of generality, we take SimCLR as a representative SSL method for evaluation. The results are shown in **Figure 6**. First, our proposed method only modifies the mini-batch construction process during the training phase of SimCLR. Even though we train a VAE, it does not affect other components of SimCLR's training pipeline, including the training objective, network architecture, and optimization algorithm. Second, training a VAE independently on ImageNet and using its feature extractor for evaluation yields an accuracy of 35.4%, in contrast to SimCLR's 70.2%. We then use the parameters of the VAE's feature extractor to initialize the feature encoder of SimCLR and retrain SimCLR from this initialization. The resulting accuracy is 68.7%, which is 1.4% lower than that of SimCLR trained from scratch. In comparison, SimCLR combined with our method achieves an accuracy of 73.3%. These results demonstrate the fairness of our evaluation and confirm that the performance gain is not due to the additional VAE.

## C.2. Evaluation on More Modalities

The proposed method can be applied in various fields and domains, e.g., instance segmentation, video tracking, sample generation, etc., as mentioned before. Here, we provide the experiments of the proposed method on text modality-based datasets, i.e., IC03 and IIIT5K (Yasmeen et al., 2020), which we have conducted before. We follow the same experimental settings as mentioned in (Aberdam et al., 2021). The results shown in **Table** 8 demonstrate that the proposed method achieves stable effectiveness and robustness in various modalities combined with the above experiments.

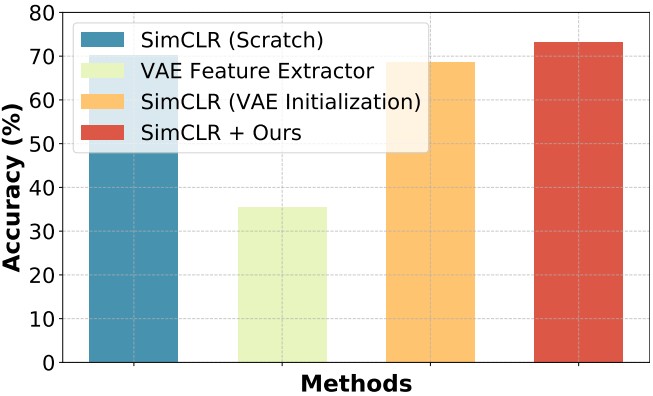

Figure 6: Evaluation for the VAE-based components.

Table 10: Performance comparison on PACS dataset.

| Method | Photo | Sketch | Cartoon | Painting (Unseen) | Average |
|---|---|---|---|---|---|
| SimCLR | 86.4 | 85.1 | 87.2 | 74.3 | 80.7 |
| SimCLR+Ours | 88.0 | 87.4 | 90.1 | 79.2 | 85.0 |
| BYOL | 83.9 | 84.6 | 82.7 | 64.5 | 74.2 |
| BYOL+Ours | 84.2 | 86.9 | 85.0 | 70.8 | 78.9 |

Table 11: Performance comparison on OfficeHome dataset

| Method | A $\rightarrow$ C | A $\rightarrow$ P | A $\rightarrow$ R | C $\rightarrow$ A | C $\rightarrow$ P | C $\rightarrow$ R | P $\rightarrow$ A | P $\rightarrow$ C | P $\rightarrow$ R | R $\rightarrow$ A | R $\rightarrow$ C | R $\rightarrow$ P |
|---|---|---|---|---|---|---|---|---|---|---|---|---|
| SimCLR | 58.2 | 63.5 | 69.8 | 78.9 | 69.7 | 66.8 | 63.4 | 52.3 | 58.4 | 56.1 | 72.9 | 71.0 |
| SimCLR+Ours | 61.1 | 65.2 | 71.9 | 81.1 | 72.0 | 68.2 | 67.5 | 59.1 | 59.9 | 61.2 | 74.8 | 73.5 |

## C.3. Evaluation on OOD Tasks

In addition to validating the proposed method on standard and few-shot transfer learning scenarios, we also specifically test it on benchmark datasets targeting the out-of-distribution (OOD) problem, including PACS, OfficeHome, Waterbirds, and ColoredMNIST. Specifically, we evaluate the performance of SSL baselines before and after introducing the proposed PID on these three datasets. For PACS, we follow the experimental setup in **Section 5.2** to evaluate the most commonly used SSL baselines (i.e., SimCLR and BYOL) on three domains (i.e., Photo, Sketch, and Cartoon) training on these domains and testing on all four domains, including Photo, Art, Cartoon, and Sketch, as well as the average performance. The results are shown in **Table 10**. For OfficeHome, we randomly select one domain as the source domain for training and another as the target domain. The labels for the source domain are predefined, whereas the labels for all target domains are unknown. We then evaluate the performance change of SimCLR before and after introducing PID, with results shown in **Table 11**. For the Waterbirds dataset, we adopt the implementation from (Zare & Van Nguyen, 2023). During training, waterbirds (landbirds) are predominantly paired with water (land) backgrounds. However, at test time, the distribution of backgrounds is altered, creating a domain shift. In our evaluation, we report both the average performance and the performance of the worst-performing group. The results are shown in **Table 12**. From the results, we can observe that our method consistently yields improvements, with particularly significant gains observed for the worst-performing group, indicating the effectiveness of our approach in addressing domain shifts. Finally, for the Colored-MNIST dataset, we follow the experimental settings mentioned in (Huang et al., 2024). In the benchmark, the task involves classifying 10-digit classes, where 10% of the labels are randomly reassigned to evaluate the model's robustness to label noise. During training, images belonging to class '0' (or '1') are colored red (or green) with a probability of 77.5%, and another random color with a probability of 22.5%. At test time, the color scheme is reversed, allowing us to assess the model's reliance on color cues for classification. The results shown in **Table 13** demonstrate that our method enhances out-of-distribution (OOD) test accuracy by nearly 10%, highlighting the improvement in the model's generalization ability.

## D. Related Works for Spurious Correlation

In the recent work on SSL, there has been growing interest in understanding its vulnerability to spurious correlations (Hamidieh et al., 2024; Wang et al., 2022; 2023a). These correlations arise when models learn associations from data that do not truly reflect the underlying causal structure, but instead are coincidental or context-specific patterns (Pearl, 2009). This susceptibility can undermine the effectiveness of SSL, particularly when dealing with diverse data environments.

Some works have been proposed to alleviate the effects of spurious correlations in SSL. Hamidieh et al. (Hamidieh et al., 2024) introduced a method that counteracts these correlations by expanding the feature space, thereby providing more diverse training views to mitigate misleading associations. Park et al. (Park et al., 2024) proposed that spuriously correlated attributes make neural networks inductively biased towards encoding lower effective rank representations and used rank regularization to eliminate biased samples. Another notable contribution comes from Chen et al.(Zhu et al., 2023), who explored the use of a data reweighting strategy to reduce the importance of data samples that may contain spurious correlations. These methods attempt to eliminate spurious correlations by filtering or enhancing SSL samples at the sample level. Although this approach has proven effective by excluding samples that may contain spurious correlations, it is difficult to ensure that the learned features are still reliable due to the partial unobservability of spurious correlations and

Table 12: Performance comparison on Waterbirds dataset.

| Method | Test Accuracy | Worst Group |
|---|---|---|
| SimCLR | 76.2 | 19.2 |
| SimCLR + Ours | 78.0 | 24.9 |
| MAE | 74.9 | 17.6 |
| MAE + Ours | 77.2 | 22.1 |

Table 13: Performance comparison on ColoredMNIST dataset.

| Method | Accuracy(%) |
|---|---|
| SimCLR | 12.7 |
| SimCLR + Ours | 23.5 |
| MAE | 15.1 |
| MAE + Ours | 24.9 |

variable coupling. In contrast, our work directly addresses the impact that spurious correlations might cause, utilizing the independence between unobserved variables and anchors under post-intervention distributions to ensure the reliability of the learned representations.

# E. Task Distribution & Data Distribution

**Task Distribution**: Task distribution refers to a set of tasks and their underlying distribution, where each task has its own specific objectives and associated data distribution. It is often used in meta-learning or multi-task learning scenarios to describe the diversity and variation across tasks.

For example, in a meta-learning scenario, the task distribution could include:

- A "cat vs dog" classification task (Task 1).

- A "car vs airplane" classification task (Task 2).

- A "bird vs fish" classification task (Task 3).

These tasks form the task distribution, and the meta-learning model is trained across this task space.

**Data Distribution**: Data distribution refers to the statistical distribution of data samples within a single task, typically described as the joint distribution $P(X, Y)$ of input $X$ and labels $Y$.

Task distribution describes the variability between tasks in a learning system, focusing on generalization across tasks. Data distribution focuses on the variability within a single task, addressing adaptation to specific data characteristics. The two concepts are hierarchical: task distribution governs the diversity of tasks, while each task has its own distinct data distribution.

**Reformulation of OOD Generalization as Generalization on Task Distributions**: We organize the whole process into the following steps:

**Step 1**: First, we provide the formal definition of task distribution.

Without loss of generality, let us use a classification task as an example. We define $X_{tr}^a = \{(x_i^a, y_i^a)\}_{i=1}^N$ as a training dataset, where $x_i^a$ represents a sample, $y_i^a$ represents the corresponding label, $a$ denotes the dataset index, and $N$ denotes the number of samples in the dataset. For a classification task, the goal is to learn a classifier $p^a(y_i^a|x_i^a)$, so that for any given sample $x_i^a$, the corresponding label can be predicted.

If $N \to +\infty$, $X_{tr}^a$ can be approximated as containing all the information necessary for the classification task and can thus be regarded as a complete dataset for a classification task. Simply put, the elements of a classification task include: the

classifier and the dataset. We denote a task as $(X_{tr}^a, p^a(y_i^a|x_i^a))$. Then, the discrete distribution of tasks can be expressed as $\{X_{tr}^a, p^a(y_i^a|x_i^a)\}_{a=1}^M$, where $M$ represents the number of tasks.

Furthermore, when $a$ is different, the label space corresponding to $y_i^a$ is also different. For example, when $a = 1$, the label space is $\{Cat, Dog\}$, and when $a = 2$, the label space is $\{Plane, Train\}$. If $M \to +\infty$, $\{X_{tr}^a, p^a(y_i^a|x_i^a)\}_{a=1}^M$ can be regarded as a complete task distribution.

**Step 2**: Next, we reformulate SSL from the perspective of task distribution.

In Section 2 and Section 3.1, we explain why a mini-batch in SSL can be viewed as a task. Simply put, for a given mini-batch, it can be expressed as: $X_{tr,a}^{aug} = \{x_a^i, x_{anchor,a}^i\}_{i=1}^{2N}$, where $N$ denotes the number of ancestor samples in the mini-batch, $a$ represents the index of the mini-batch, and $x_{anchor,a}^i$ can be regarded as the label of the augmented sample. Meanwhile, the classifier to be learned for each mini-batch is modeled as $p^a(x_{anchor,a}^i|x_a^i)$.

Notably, the classifiers for all tasks in SSL are learned using the same classifier, i.e., the classifiers for all tasks aim to learn $p(x_{anchor,a}^i|x_a^i)$. For example, SimCLR models the classifier using a contrastive loss, while MAE models it using the $L_2$-norm. Therefore, whether D-SSL or G-SSL is used, as $M \to +\infty$, $\{(X_{tr,a}^{aug}, p(x_{anchor,a}^i|x_a^i))\}_{a=1}^M$ can be approximated as a task distribution, where $M$ represents the number of tasks.

**Step 3**: Finally, we reformulate the OOD generalization of SSL as generalization on task distributions.

In traditional machine learning, given training data, the goal is to learn $p(y|x)$. This can be understood as modeling the data distribution $p(x, y)$ as $p(x)p(y|x)$, where $p(y|x)$ is learned from the training data and transferred to the test data distribution $p(x)$. This approach assumes that the training and test data are identically and independently distributed, i.e., $p(x_{train}) = p(x_{test}) = p(x)$, and $p(x_{train}, y_{train}) = p(x_{test}, y_{test}) = p(x, y)$. Consequently, $p(x)p^{train}(y|x) = p(x)p^{test}(y|x)$, leading to $p^{train}(y|x) = p^{test}(y|x)$.

By analogy, when each data sample is treated as a task, the corresponding learning objective becomes $p(p^a(x_{anchor,a}^i|x_a^i)|X_{tr,a}^{aug})$. This learning goal is similar to that in meta-learning [1-2], where the goal is to learn a function that can output the classifier for a given task dataset. Therefore, when the training data are drawn from a task distribution, the learning objective is to model the task distribution, i.e., to learn $p(p^i(y|x)|p(\text{task } i))$, such that it applies to both training and test tasks. Since training and test tasks are different, from the perspective of the training tasks, the test tasks represent OOD scenarios. However, from the perspective of the task distribution, both training and test tasks belong to the same task distribution.

Thus, from the viewpoint of traditional machine learning, SSL can be considered as training with mini-batches of size 1, where each training sample is a training task. One open problem is how to model $p(p^i(y|x)|p(\text{task } i))$. Since we define the classifier $p(x_{anchor,a}^i|x_a^i)$ for each SSL training task as identical, $p(p^i(y|x)|p(\text{task } i))$ can be directly modeled as $p(x_{anchor,a}^i|x_a^i)$, which applies to any sample from any task.

In conclusion, combining **Step 1-3**, we reformulate OOD generalization as generalization on task distributions.

## F. Intuitive Explanations for Assumption 3.3 and Theorem 4.7

**Assumption** 3.3 illustrates that regardless of whether $e \in \mathcal{D}$ or $e \in$ PID, $x^+$ is generated under the control of two variables, $s$ and $x^{\text{label}}$. Therefore, given $x^+$, $s$ and $x^{\text{label}}$ are conditionally independent, regardless of the correlation between them.

From **Assumption** 3.3, the optimal $f$ should be $F_{x^{\text{label}}}$. However, without additional constraints, it is difficult to obtain this optimal $f$. **Theorem** 3.4 provides a way to obtain another good $f$, defined as $f^*$ in the theorem. Why is $f^*$ considered good? This is because **Theorem** 3.4 implies that when $\mathcal{D}$ is sufficiently large and diverse, an optimal $f^*$ trained on one distribution will perform worse than random guessing in some other environment. Under such conditions, no other $f$ obtained from training on any distribution can achieve better worst-case OOD performance than the PID. Why is focusing on the worst-case scenario better than other cases? During training, we minimize the worst-case scenario, which involves minimizing: $\max_{e \in \mathcal{D}} \mathcal{L}^e(p_f(x^{\text{label}}|x^+))..$ For any $f$, the term $\max_{e \in \mathcal{D}} \mathcal{L}^e(p_f(x^{\text{label}}|x^+))$ is always greater than or equal to $\mathcal{L}^e(p_f(x^{\text{label}}|x^+))$ for any specific environment $e$. If we learn an $f$ that minimizes the worst-case term $\max_{e \in \mathcal{D}} \mathcal{L}^e(p_f(x^{\text{label}}|x^+))$, then we naturally minimize $\mathcal{L}^e(p_f(x^{\text{label}}|x^+))$ for all $e$ in $\mathcal{D}$. This ensures robustness across all scenarios, making the worst-case optimization strategy effective for improving OOD performance.

The high-level explanation of **Theorem** 4.7 can be presented as follows: 1) From **Definition** 4.4, it follows that if $ba(s)$

can be identified, then $s$ and $x^{\text{label}}$ are conditionally independent given $ba(s)$; 2) In this paper, $ba(s)$ is implemented as described in Equation (5) in the main text. The key challenge lies in obtaining $s$. As shown in Section 4.1, we explain the identifiability of $s$, as well as how each label is modeled using a distribution for $s$. During implementation, we sample from this distribution to generate a series of discrete vectors that approximate $s$ associated with a specific label; 3) From Equation (2) in the main text, we have: $p(x^+, x^{\text{label}}, s) = p(x^+|x^{\text{label}}, s)p(x^{\text{label}})p(s|x^{\text{label}})$. If we select sample pairs for a mini-batch such that all pairs share the same $ba(s)$, the resulting mini-batch can be considered as constructed under the same $ba(s)$. In other words, the samples in the mini-batch are conditioned on $ba(s)$. Combined with the argument in **Point 1)**, we have: $p(x^+, x^{\text{label}}, s) = p(x^+|x^{\text{label}}, s)p(x^{\text{label}})p(s)$, which ensures that the mini-batch satisfies PID. The key to achieving PID is ensuring that all sampled examples in the mini-batch have consistent $ba(s)$, i.e., the background information is the same. This allows SSL training to focus on foreground information while disregarding background information.

## G. More Explanation for the Identifiability of Spurious Variable

To better address the identifiability of spurious variables in the context of SSL, we organize the response into the following steps:

**Step 1**: First, we need to clarify that in Section 2 and Section 3.1, we propose a new perspective for understanding SSL. Taking classification as an example, under this new perspective, each mini-batch during the training phase of SSL can be treated as an independent multi-classification task. Different mini-batches correspond to different classification tasks. In contrast, the traditional perspective of SSL considers the entire dataset as a single task for unsupervised learning.

Therefore, under the new perspective, the training samples in each mini-batch can be considered labeled. Whether these labels are accurate is not our concern for now, as this falls under the domain of Bayesian error. Consequently, in this sense, spurious variables can be identifiable.

**Step 2**: We first explain what we mean by the distribution of tasks, using classification as an example. A learning task can be narrowly defined as assigning a label to each sample in a dataset, where the label types are finite. This dataset can represent the task, and the entirety of the dataset can be regarded as the data distribution of that task. Thus, different tasks correspond to different datasets, with distinct label types (tasks with the same label types are considered the same). In this way, a task distribution is essentially the distribution over these datasets, with each element of the task distribution corresponding to a specific dataset.

**Step 3**: Next, we point out that in this paper, the spurious variable $s$ indeed takes values in an infinite space since it is represented by a high-dimensional vector. The values of this vector can be arbitrary. We must define $s$ as taking infinite values because, as discussed in Section 2 and Section 3.1, we reinterpret SSL as learning a task distribution where the label types involved are infinite.

Different labels may correspond to different latent variables. These differences are represented by different distributions, i.e., we model the distribution $q_\phi(s|x^+, x^{\text{label}})$ using a latent variable model. This allows us to derive the distribution of the spurious variable $s$ for any given label. The values of the probability density can be understood as the degree of correlation between a specific label and a particular value of the latent variable. Hence, given a label, once its conditional distribution $q_\phi(s|x^+, x^{\text{label}})$ is determined, we can estimate the corresponding spurious variable $s$ through sampling.

**Step 4**: We do not theoretically prove that the latent variable model can directly identify the spurious variable $s$. In this paper, the identification of $s$ is based on a strong assumption—**Assumption** 4.1 in the paper. This assumption is justified as follows:

Based on the literature (Blei et al., 2017; Sriperumbudur et al., 2013), which expresses the true prior in closed form, we deduce that when the causal relationship between the latent covariate and the label changes with the tasks, an exponential family distribution is capable of modeling the conditional distribution $p(s|x^{\text{label}})$.

Combining **Step 1**, **Step 2**, and **Step 3**, we satisfy the condition that the causal relationship between the latent covariate and the label changes with the tasks.

**Step 5**: **Theorem** 4.3 is also based on **Assumption** 4.1. The key result of **Theorem** 4.3 is that we can uniquely identify $\phi$ and $(f, g, A)$. However, this strong assumption imposes certain limitations on the accuracy of spurious variable identification, which is a topic for future research. Despite this strong assumption, our experimental results demonstrate the effectiveness

of our method.

## H. Explanation on Why Worst-Case is Good for OOD Generalization in SSL

During training, we minimize the loss in the worst-case scenario: $\max_{e \in \mathcal{D}} \mathcal{L}^e(p_f(x^{\text{label}}|x^+))$. For any $f$, the worst-case loss $\max_{e \in \mathcal{D}} \mathcal{L}^e(p_f(x^{\text{label}}|x^+))$ is always greater than or equal to the loss in any specific environment $e$: $\mathcal{L}^e(p_f(x^{\text{label}}|x^+))$.

If we learn an $f$ that minimizes the worst-case loss $\max_{e \in \mathcal{D}} \mathcal{L}^e(p_f(x^{\text{label}}|x^+))$, then $f$ naturally minimizes $\mathcal{L}^e(p_f(x^{\text{label}}|x^+))$ for all $e$ in $\mathcal{D}$. This ensures robust performance across all scenarios, making the worst-case optimization strategy particularly effective for enhancing OOD generalization.

Therefore, the worst-case can help us to better learning $p_f(x^{\text{label}}|x^+)$. A better $p_f(x^{\text{label}}|x^+)$ can achieve a less empirical risk in the training distribution. As shown in **Step 3** of "**Reformulation of OOD Generalization as Generalization on Task Distributions**" in **Appendix** E, from the viewpoint of traditional machine learning, SSL can be considered as training with mini-batches of size 1, where each training sample is a training task. Then, from PAC theory, we can obtain that minimizing the empirical risk can lead to minimize expected risk, and the expected risk is calculated based on a series of test tasks. Thus, the worst-case can improve the OOD generalization in SSL.

## I. The Logical Structure and Viewpoint

To make it easier for readers to understand, we further clarify the logical structure and viewpoints of this paper in the following steps:

**Step 1: Reformulate SSL from the perspective of task distribution.** From the perspective of task distribution, SSL can be understood as learning a distribution of tasks. Each task during training is a classification task: for G-SSL, the classifier is modeled using the $L_2$-norm, while for D-SSL, the classifier is modeled using contrastive loss. It is crucial to emphasize that we unify the concepts of alignment, classifier, and loss function, as they are essentially the same in our formulation. For further details, please refer to "**Reformulation of OOD Generalization as Generalization on Task Distributions**" in Appendix E.

**Step 2: Model the learning process of each task using a fuzzy SCM.** In our new understanding, we model the learning process of each task using a fuzzy SCM. It is considered "fuzzy" because the relationship between $s$ and $x^{\text{label}}$ is unclear, as shown in Figure 1. We elaborate on this in Section 3.1, explaining that the relationship between $s$ and $x^{\text{label}}$ varies across tasks and cannot be captured using a single SCM.

**Step 3: Explain how spurious variables impact OOD generalization in SSL.** In Section 3.1, we explain that under our new understanding, current SSL methods may learn spurious variables, which negatively impact OOD generalization performance. Specifically, spurious variables affect OOD generalization as follows:

- Spurious variables make it challenging to learn each specific task properly.

- This, in turn, hinders the modeling of the task distribution.

- Consequently, SSL performs poorly on test tasks, where test tasks differ from training tasks.

- This ultimately undermines OOD generalization.

**Step 4: Core argument — improving SSL's OOD performance despite spurious variables.** At this point, the core argument of this paper emerges. Through **Theorem** 3.4, we demonstrate that even in the presence of spurious variables, it is possible to propose a method to enhance OOD generalization. From **Assumption** 3.3, the optimal $f^*$ should be $F_{x^{\text{label}}}$. However, without additional constraints, obtaining the optimal $f^*$ is challenging. **Theorem** 3.4 implies that when $\mathcal{D}$ is sufficiently large and diverse, an optimal $f^*$ trained on one distribution will perform worse than random guessing in some other environments. Under such conditions, no other $f$ obtained from training on any distribution can achieve better worst-case OOD performance than the PID. In other words, even in the presence of spurious variables, e.g. G-SSL, our proposed approach can improve the OOD performance of SSL.

**Step 5: Why is the worst-case scenario critical for improving SSL's OOD performance?** This is a key insight into how our approach improves SSL's OOD performance. During training, we minimize the loss in the worst-case

scenario: $\max_{e \in \mathcal{D}} \mathcal{L}^e(p_f(x^{\text{label}}|x^+))$. For any $f$, the worst-case loss $\max_{e \in \mathcal{D}} \mathcal{L}^e(p_f(x^{\text{label}}|x^+))$ is always greater than or equal to the loss in any specific environment $e$: $\mathcal{L}^e(p_f(x^{\text{label}}|x^+))$. If we learn an $f$ that minimizes the worst-case loss $\max_{e \in \mathcal{D}} \mathcal{L}^e(p_f(x^{\text{label}}|x^+))$, then $f$ naturally minimizes $\mathcal{L}^e(p_f(x^{\text{label}}|x^+))$ for all $e$ in $\mathcal{D}$. This ensures robust performance across all scenarios, making the worst-case optimization strategy particularly effective for enhancing OOD generalization.

**Step 6: The proposed approach — achieving PID.** Our approach achieves PID based on SCM. Why does **Algorithm** 1 achieve PID? A high level explanation is presented in "**Intuitive Explanations for Theorem** 4.7" in **Appendix** F.

