# OpenReview forum: "On the Out-of-Distribution Generalization of Self-Supervised Learning"
_ICML.cc/2025/Conference — ICML 2025 poster_

### Official Review · Reviewer_fDza · 2025-03-11

**Overall Recommendation:** 3

**Summary:**

This paper focuses on the out-of-distribution generalization of self-supervised learning. The authors first give one plausible explanation for SSL having OOD generalization, then analyze and conclude that SSL learns spurious correlations during the training process from the perspective of generation and causal inference. To address this issue, they further propose a post-intervention distribution (PID) grounded in the Structural Causal Model. Experiments verify the advantages of their method.

## Update After Rebuttal
Thank you for addressing my concerns. I maintain my positive score.

**Claims And Evidence:**

It seems convincing.

**Essential References Not Discussed:**

The essential references seem sufficient.

**Experimental Designs Or Analyses:**

I have checked the experimental designs and analyses, and the experimental results are impressive.

**Methods And Evaluation Criteria:**

The method and evaluation criteria make sense.

**Other Comments Or Suggestions:**

See weaknesses (above) and questions (below).

**Other Strengths And Weaknesses:**

Strengths:
1. This article provides a wealth of theoretical analysis, making the entire work more solid.
2. The experimental results are impressive.


Weaknesses:
1. While the theoretical aspects are robust, the practical implementation of these concepts, especially the integration of causal inference in SSL, might be complex and computationally intensive. This could limit its applicability in environments with constrained computational resources.
2. Some of the causal assumptions made may not hold in all real-world scenarios, which could affect the generalizability of the findings. A deeper exploration of these assumptions, including conditions under which they may not be valid, would provide a more comprehensive view of the method’s applicability.
3. Some key terms and variables used throughout the paper could be defined more clearly to avoid ambiguity, enhancing the paper’s accessibility to a broader audience.

**Questions For Authors:**

1. Could you provide more detailed insights into how the Post-Intervention Distribution (PID) is specifically integrated into the self-supervised learning process? It would be beneficial to understand the operational steps or algorithms used to enforce PID constraints during mini-batch preparation.
2. You mentioned that the proposed method minimizes spurious correlations. Can you discuss any specific metrics or evaluation criteria used to measure the extent of spurious correlations before and after applying your method?
3. Could you discuss the scalability of your proposed method, particularly in terms of computational resources and time required as the dataset size increases? Is the method feasible for large-scale real-world applications where computational efficiency is critical?

**Relation To Broader Scientific Literature:**

By analyzing the mini-batch construction during the SSL training phase, this paper gives one plausible explanation for self-supervised learning (SSL) having OOD generalization. Moreover, this paper also analyzes and concludes that SSL learns spurious correlations during the training process, which leads to a reduction in OOD generalization.

**Theoretical Claims:**

I have checked the proofs of theoretical claims, and the entire theoretical derivation and claims seem appropriate.

---

> ### Author Rebuttal · Authors · 2025-03-31
>
> **Response to Weaknesses 1 & Questions 3**:
>
> Thank you for pointing these out. The proposed method has two main phases with the following complexity analysis per mini-batch (batch size $B$, dataset size $D$):
>
> **Step 1: Latent Variable Model Training:**
>    - **$q_\phi(s|x^+,x^{\rm label})$:** Each sample requires a forward pass with cost $O(C_\phi)$, totaling $O(B \cdot C_\phi)$.
>    - **$p_f(x^+|s,x^{\rm label})$:** Each sample incurs a cost $O(C_f)$, totaling $O(B \cdot C_f)$.
>    - **$g$ for $\lambda^e$:** Computed once per mini-batch with cost $O(C_g)$.
>    - **KL-Divergence:** Involves operations over the latent dimension $n$ and sufficient statistic dimension $k$, contributing $O(B\cdot n\cdot k)$.
>    - **Orthogonality Regularization:** Requires $O(n\cdot k^2)$, which is constant when $n$ and $k$ are small.
>
>    Thus, the training phase complexity is approximately:   $
>    O\Big(B\cdot (C_\phi + C_f + n\cdot k) + C_g + n\cdot k^2\Big).
>    $
>
> **Step 2:  Algorithm 1**
>    - **Propensity Score Calculation:** For each sample, computing scores across the $D$ candidates costs $O(D\cdot n\cdot k)$, leading to a total of $O(D^2\cdot n\cdot k)$ for the mini-batch.
>    - **Matching Operation:** A brute-force matching over $D$ samples yields an additional $O(D^2)$.
>
>    Therefore, the sampling phase has an overall complexity of approximately:  $
>    O(D^2\cdot n\cdot k).
>    $
>
> **Step 3: Overall Complexity**
>
> The combined complexity per mini-batch is: $
> O\Big(B\cdot (C_\phi + C_f + n\cdot k) + C_g + n\cdot k^2 + D^2\cdot n\cdot k\Big).
> $
>
> The symbols $C_\phi$, $C_f$, and $C_g$ represent the computational cost for a single forward pass (or operation) of each respective network module. For specific computational resources and time, please refer to **Response to Weaknesses 2** of **Rebuttal for Reviewer 1Q1V**
>
> ----
>
> **Response to Weaknesses 2**
>
> Thank you for pointing this out. We provide a deeper exploration of **Assumption 3.3** and **Assumption 4.1** in our response.
>
> For **Assumption 3.3**, we implicitly assume that the noise is independent of both $ s $ and $ x^{\rm label} $. However, in many practical scenarios, noise may be correlated with either the latent variables or the observed features—for example, sensor noise that correlates with lighting conditions in image data—which can interfere with the separation between causal and non-causal factors.
>
> Regarding **Assumption 4.1**, the main concern lies in the potential mismatch between the true conditional distribution in real-world data and the assumed exponential family. If the actual distribution is more complex or exhibits behaviors that go beyond this family—such as multi-label or multi-instance characteristics—then the applicability of our method may be compromised.
>
> ----
>
> **Response to Weaknesses 3**:
>
> Thank you for pointing this out. In the final version, we will add a table to illustrate all terms and variables related to our method.
>
> ----
>
> **Response to Questions 1**:
>
> Thank you for pointing this out. We explain this issue through the following steps:
>
> **Step 1: How do we implement PID**
>
> According to *Definition 4.4* in the original submission, $s$ and $x^{\rm label}$ are conditionally independent given $ba(s)$. Based on this, if all pairs in a mini-batch share the same $ba(s)$, then within this mini-batch, $s$ and $ x^{\rm label} $ can be considered independent. Consequently, such a mini-batch can be viewed as being sampled from a **PID**.
>
> **Step 2: How is this integrated into SSL**
>
> In the training phase of SSL, a mini-batch is typically sampled from the training data prior to each iteration. In standard SSL, this mini-batch is randomly sampled. In contrast, our method constructs the mini-batch using **Algorithm 1** from the original submission. That is, our approach embeds into SSL by replacing the mini-batch sampling process with **Algorithm 1**, without altering any other part of the SSL training procedure.
>
> According to **Algorithm 1**, the core criterion for selecting samples is to ensure that the $ ba(s) $ values of each pair are as similar as possible. This ensures that the resulting mini-batch has consistent $ ba(s) $ across all samples, thereby forming a PID.
>
> ----
>
> **Response to Questions 2**:
>
> Thank you for pointing this out. Instead of proposing specific metrics or evaluation criteria, we run a toy experiment on the COCO dataset [1] with two different experimental settings: 1) training and testing the SSL model on full images; 2) training and testing the SSL model on foreground images. Setting 2) can be thought of as not being subject to background semantic confounding. For the Top 1  classification accuracy, the results of  SimCLR are 39.66 and 50.19, the results of  SimCLR + Ours are 45.25 and 51.48. We observe that our method gives closer results in both settings and significantly outperforms SimCLR. Thus, it can be concluded that our method learns less spurious correlations.
>
> [1] Microsoft coco: Common objects in context. ECCV, 2014.

---

### Official Review · Reviewer_vxgt · 2025-03-12

**Overall Recommendation:** 4

**Summary:**

This work propose a minibatch sampling strategy to select pairs of samples in the mini-batch to enhance the OOD geralization ability of SSL methods. By investigating on a causal perspective from the constructed SCM model, the method propose a Post-Intervention Distribution, which can be realized by balancing score.

**Claims And Evidence:**

I do not find any evident errors in the claims.

**Essential References Not Discussed:**

No.

**Experimental Designs Or Analyses:**

This work primarily addresses the OOD generalizability of SSL methods; however, the experiments do not include any OOD datasets, such as _Waterbirds and CMNIST_. Conducting experiments directly on OOD datasets would help evaluate the effectiveness of the proposed sampling strategy.

**Methods And Evaluation Criteria:**

The method seems convincing, essentially when the balancing condition holds, there is no changes of distribution in spurious features, therefore the SSL method will focus on the invariant features.

**Other Comments Or Suggestions:**

N/A

**Other Strengths And Weaknesses:**

1. The biggest concern for me is the experiment setting, i.e., it does not involve any OOD dataset for evaluation, however, the main goal of this work is to enhance OOD generalization ability of SSL methods.

**Questions For Authors:**

1. Why using a exponential family distribution to model $p(s \mid x^{label})$? Why not using a reversible neural nets which may achieve higher expressibity and simplify the design?

**Relation To Broader Scientific Literature:**

This work relates to the self-supervised learning literatures and domain generalization, as well as causal inference literatures.

**Theoretical Claims:**

The results of Theorem 3.4 should be correct that the loss minimizes the worst-case risk, this is a well-defined target in invariant learning literatures. I am not sure if Theorem 4.3 is correct as I am not familiar with the identifiability theory.

---

> ### Author Rebuttal · Authors · 2025-03-31
>
> **Response to Weaknesses 1 & Experimental Designs Or Analyses**:
>
> Thank you for pointing this out. We clarify this issue through the following steps:
>
> **Step 1: How the original submission constructs the OOD task**
>
> The transfer learning task and the few-shot learning task can be regarded as OOD (out-of-distribution) tasks, as the training and test datasets in these tasks follow different data distributions. Meanwhile, in **Appendix C.3** of the original submission, we also provide evaluation results on two OOD datasets, namely the Colored-MNIST dataset and the PACS dataset.
>
> **Step 2: Results on Waterbirds dataset and CMNIST dataset**
>
> For Waterbirds, we follow the implementation in ZARE et al. (2023) "Evaluating and Improving Domain Invariance in Contrastive Self-Supervised Learning by Extrapolating the Loss Function". During training, waterbirds (landbirds) predominantly appear on water (land) backgrounds; however, the distribution is altered at test time. We report both the average and worst-group performance. The results in the table below demonstrate that our method yields consistent improvements, particularly enhancing performance for the worst-performing group.
>
> | Method      | Test Accuracy | Worst Group |
> | ----------- | ------------- | ----------- |
> | SimCLR      | 76.2          | 19.2        |
> | SimCLR+Ours | 78.0          | 24.9        |
> | MAE         | 74.9          | 17.6        |
> | MAE+Ours    | 77.2          | 22.1        |
>
> For Colored-MNIST, we follow the implementation in Huang et al. (2024) "On the Comparison between Multi-modal and
> Single-modal Contrastive Learning". The task is a 10-class digit classification, with 10% of the labels randomly reassigned. During training, images belonging to class ‘0’ (or ‘1’) are colored red (or green) with a probability of 77.5%, and another random color with a probability of 22.5%. For the test set, the coloring scheme is reversed relative to training, which allows us to evaluate the extent to which the model relies on color cues for classification. The results in the table below show that our method improves OOD test accuracy by nearly 10%.
>
> | Method      | Test Accuracy |
> | ----------- | ------------- |
> | SimCLR      | 12.7          |
> | SimCLR+Ours | 23.5          |
> | MAE         | 15.1          |
> | MAE+Ours    | 24.9          |
>
> These results further demonstrate that our proposed method effectively improves the OOD generalization performance of SSL.
>
> ----
>
> **Response to Question 2**:
>
> Thank you for pointing this out. According to Assumption 3.3 in the original submission, both $ x^{\rm label} $ and $s$ can be obtained through an invertible neural network. Training such an invertible neural network typically requires training data in the form of $(x^{\rm label} _i, s _i, x _i^+) _{i=1}^N $. However, we did not adopt this mechanism directly because we did not have access to such training data. In particular, we were unable to provide the corresponding $ s_i $ for each pair. Therefore, we opted to use a **Learning Latent Variable Model** approach instead.

---

### Official Review · Reviewer_1Q1V · 2025-03-12

**Overall Recommendation:** 2

**Summary:**

The paper explores the **out-of-distribution (OOD) generalization** of self-supervised learning (SSL). It analyzes how mini-batch construction in SSL training influences OOD generalization and argues that SSL models often learn **spurious correlations**, which hinder their ability to generalize to unseen distributions. To address this issue, the paper introduces a **post-intervention distribution (PID)** based on **Structural Causal Models (SCMs)**. This ensures that spurious variables and label variables remain independent, improving OOD generalization.

Furthermore, the authors propose a **mini-batch sampling strategy** that enforces PID constraints through a latent variable model. They provide theoretical proof of the identifiability of their method and validate it with empirical results. Experiments on various downstream OOD tasks demonstrate that their approach significantly enhances SSL’s generalization performance.

### **Strengths:**
1. **Novel Causal Perspective on SSL OOD Generalization**
   - The paper offers a compelling causal analysis of why SSL struggles with OOD generalization and how spurious correlations arise.

2. **Innovative Mini-Batch Sampling Strategy**
   - Unlike traditional batch sampling, the method ensures **spurious correlations are minimized**, leading to better OOD generalization.

3. **Strong Empirical Performance**
   - The proposed method consistently improves performance across diverse benchmarks, including **unsupervised, semi-supervised, transfer learning, and few-shot learning tasks**.

**Claims And Evidence:**

The claims in the paper are partially supported by the evidence in the experiment part. However, the reviewer is concerned about the lack of evaluation on mask-autoencoder based pre-training methods like [1,2] in the main paper. Although some results are provided in the supplementary, it can be worthwhile to add the comparison in the experiments in the main paper together with contrastive-based methods, as in the analysis and proof part, the authors formulate discriminated-based and generative-based methods with a unified framework.

[1] He, K., Chen, X., Xie, S., Li, Y., Doll´ ar, P., and Girshick, R. Masked autoencoders are scalable vision learners. In Proceedings of the IEEE/CVF conference on computer vision and pattern recognition, pp. 16000–16009, 2022.
[2] Tong, Z., Song, Y., Wang, J., and Wang, L. Videomae: Masked autoencoders are data-efficient learners for self-supervised video pre-training. Advances in neural information processing systems, 35:10078–10093, 2022.

**Essential References Not Discussed:**

The reviewer does not come up with essential related literature that is not discussed.

**Experimental Designs Or Analyses:**

The reviewer thinks that experimental designs are valid.

**Methods And Evaluation Criteria:**

The reviewer thinks the evaluation setting and metrics make sense for the claims.

**Other Comments Or Suggestions:**

Please refer to Other Strengths And Weaknesses part.

**Other Strengths And Weaknesses:**

1. Starting from Line 162, left column, the authors propose an assumption that "the semantic information within x+ is related only to xlabel, that is, s does not contain any causal semantics related to the task.". They provide two examples about this assumption. However, as natural images are not restricted to numbers/styles discussed in the two example, the reviewer is concerned about the assumption. More examples in ImageNet should be provided against it.
2. As additional training is required, the reviewer is concerned about the training efficiency of the proposed method. The authors should provide evaluation on it.
3. As more parameters (VAE) are introduced in the proposed method, the reviewer is concerned about the fairness of comparison. Some discussion should be provided.

**Questions For Authors:**

Please refer to Other Strengths And Weaknesses part.

**Relation To Broader Scientific Literature:**

The paper makes a theoretical and practical contribution by introducing a causal approach to improving OOD generalization in SSL. While the method is empirically validated, its assumptions, computational cost, and feasibility in large-scale applications could be explored further.

**Theoretical Claims:**

The reviewer checked partially of the proofs. Specifically, theoretical claims in Section 3 and 4.1 are checked.

---

> ### Author Rebuttal · Authors · 2025-03-31
>
> **Response to Claims And Evidence**:
>
> Thank you for pointing this out. In **Appendix C.1**, we report the results of MAE. Now, we present the results of VideoMAE.
>
> We transfer the learned VideoMAE + Ours on Kinetics-400 [1] to downstream action detection dataset AVA [2]. Following the standard setting [3], we evaluate on top 60 common classes with mean Average Precision (mAP) as the metric under IoU threshold of 0.5. The result is shown as follows:
>
> | Method            | Backbone      | Pre-train Dataset | Extra Labels  | $T \times \tau $ | GFLOPs | Param | mAP  |
> | ----------------- | ------------- | ----------------- | ------------- | ---------------- | ------ | ----- | ---- |
> | VideoMAE          | ViT-L         | Kinetics-700      | No            | $16 \times 4 $   | 597    | 305   | 36.1 |
> | VideoMAE          | ViT-L         | Kinetics-700      | Yes           | $16 \times 4 $   | 597    | 305   | 39.3 |
> | VideoMAE + Ours   | ViT-L         | Kinetics-700      | No            | $16 \times 4 $   | 597    | 305   | 38.7 |
> | VideoMAE + Ours   | ViT-L         | Kinetics-700      | Yes           | $16 \times 4 $   | 597    | 305   | 42.1 |
>
> In the above table, "Extra Labels" denotes that if the pre-trained models are additionally fine-tuned on the pre-training dataset with labels before transferred to AVA. $T \times \tau $ refers to frame number and corresponding sample rate. In the final version, we will add these results to the main body of our submission.
>
> [1] The kinetics human action video dataset. arXiv preprint, 2017.
>
> [2] Ava: A video dataset of spatio-temporally localized atomic visual actions. CVPR, 2018.
>
> ----
>
> **Response to Weaknesses 1**:
>
> Thank you for pointing this out. We provide some additional ImageNet-inspired examples and explanations to address the concern:
>
> **Object vs. Background:**
>   Consider an ImageNet class like "Golden Retriever." The causal semantics for recognizing a Golden Retriever primarily reside in the dog’s shape, fur texture, and facial features. Although many images might have different backgrounds—such as parks, beaches, or urban settings—these background elements (which could be captured by $s$) are not causally responsible for the image being classified as a Golden Retriever. In this case, $x^{\rm label}$ would capture the object-specific features, while $s$ would account for non-causal variations like the background.
>
> **Intra-Class Variability:**
>   Take another class, such as "Volcano." A volcano can be pictured under different weather conditions, from different angles, and with various surrounding landscapes. While these environmental or stylistic factors vary widely, the key causal semantics—such as the volcano’s structure, cone shape, and crater—remain consistent. Again, $s$ may vary (capturing changes in lighting, weather, or background) without affecting the causal information needed to identify a volcano.
>
> ----
>
> **Response to Weaknesses 2**:
>
> Thank you for pointing this out. We provide the model efficiency and memory footprint results of the proposed method trained on 8 NVIDIA Tesla V100 GPUs.
>
> | Method      | Training Time (Hours) |          |Memory Footprint (GB) |          |
> | ----------- | --------------------- | -------- | --------------------------------- | -------- |
> |             | CIFAR-10              | ImageNet | CIFAR-10                          | ImageNet |
> | SimCLR      | 10.4                  | 101.9    | 23.3                              | 221.6    |
> | SimCLR+Ours | 12.7                  | 106.2    | 29.7                              | 230.7    |
> | MAE         | 13.8                  | 115.5    | 26.9                              | 244.9    |
> | MAE+Ours    | 16.4                  | 122.2    | 31.2                              | 252.2    |
>
> For specific computation complexity, please refer to  **Response to Weaknesses 1 & Questions 3** of **Rebuttal for Reviewer fDza**
>
> ----
>
> **Response to Weaknesses 3**:
>
> Thank you for pointing this out. To illustrate without loss of generality, we take SimCLR as a representative SSL method. First, our proposed **Algorithm 1** only modifies the mini-batch construction process during the training phase of SimCLR. Even though we train a VAE, it does not affect other components of SimCLR’s training pipeline, including the training objective, network architecture, and optimization algorithm. Second, training a VAE independently on ImageNet and using its feature extractor for evaluation yields an accuracy of **35.44%**, in contrast to SimCLR's **70.15%**. We then use the parameters of the VAE's feature extractor to initialize the feature encoder of SimCLR and retrain SimCLR from this initialization. The resulting accuracy is **68.71%**, which is **1.45%** lower than that of SimCLR trained from scratch. In comparison, SimCLR combined with our method achieves an accuracy of **73.32%**. These results demonstrate the fairness of our evaluation and confirm that the performance gain is not due to the additional VAE.

---

### Official Review · Reviewer_VtXq · 2025-03-14

**Overall Recommendation:** 3

**Summary:**

This paper explores whether self-supervised learning possesses out-of-distribution (OOD) generalization capabilities and investigates the reasons behind its potential failure. To address this, the authors propose a Post-Intervention Distribution (PID), grounded in the Structural Causal Model. PID enables accurate OOD generalization by disentangling spurious correlations between features and labels. The authors introduce a simple yet seemingly effective mini-batch resampling technique and provide a substantial number of supporting theorems. However, I find the number of theorems somewhat excessive. I suggest that the authors consolidate the most essential ones into key theorems and present them in the manuscripts for better clarity and impact.

**Claims And Evidence:**

yes

**Essential References Not Discussed:**

yes

**Experimental Designs Or Analyses:**

yes

**Methods And Evaluation Criteria:**

yes

**Other Comments Or Suggestions:**

see weakness

**Other Strengths And Weaknesses:**

Strengths:
1. Comprehensive experiments and theoretical justifications.
2. The proposed method is simple yet appears to be effective.

Weaknesses:

1. The authors should include additional OOD benchmark datasets in the main experiments, such as Colored-MNIST and PACS.

2. I find the concept of $x_{i}^{anchor}$  mentioned in line 90 somewhat confusing, particularly regarding why it can be directly transformed into the $x^{label}$ used in Equation (1).

3. How is the balancing score function specifically implemented? Is it learnable? Does it produce a scalar output?

**Questions For Authors:**

see weakness

**Relation To Broader Scientific Literature:**

This paper explores the relationship between self-supervised learning and OOD generalization.

**Theoretical Claims:**

yes

---

> ### Author Rebuttal · Authors · 2025-03-30
>
> **Response to Weaknesses 1**:
>
> Thank you for pointing this out. Due to space limitations, we reported the experimental results of Colored-MNIST and PACS in **Tables 9** and **Tables 10** in **Appendix C.3** of the original submission. In the final version, we will move these results to the main body of the paper.
>
> ----
>
> **Response to Weaknesses 2**:
>
> Thank you for pointing this out. We explain this issue through the following steps:
>
> **Step 1: How are augmented data pairs formed in SSL**
>
> In D-SSL, each sample in a mini-batch undergoes stochastic data augmentation to generate two augmented views, e.g., for ${{x_i}}$, the augmented samples can be represented as $x^1_i$ and $x^2_i$. For G-SSL, $x_i$ is first divided into multiple small blocks, with some blocks masked, and the remaining blocks reassembled into a new sample, denoted as $x^1_i$. The original sample is then referred to as $x^2_i$. Thus, the augmented dataset in SSL (whether D-SSL or G-SSL) is represented as $X_{tr}^{aug} = ( {x_i^1,x_i^2} )_{i=1}^N$. $(x_i^1,x_i^2)$ forms the $i$-th pair.
>
> The above statement can be found in the first paragraph of **Section 2** in the original submission.
>
> **Step 2: How is the anchor formed in SSL**
>
> The objective of D-SSL methods typically consists of two components: alignment and regularization. The alignment part is to maximize the similarity between samples that share the same pair in the embedding space, and the regularization part aims to constrain the learning behavior via inductive bias. It is noteworthy that “alignment” in D-SSL is often implemented based on anchor points, that is, viewing one sample in **a pair** as an anchor, the training process of such SSL methods can be seen as gradually pulling **the other sample in this pair** (a pair consists of two augmented samples) towards the anchor. Meanwhile, G-SSL can be regarded as implementing alignment of samples within a pair based on an encoding-decoding structure, by inputting sample $x^1_i$ into this structure to generate a sample, and making it as consistent as possible with sample $x^2_i$. The concept of anchor is also applicable to G-SSL, where $x^2_i$ is viewed as the anchor, and thus the training process of such SSL methods can be viewed as gradually constraining $x^1_i$ to approach $x^2_i$.
>
> The above statement can be found in the second paragraph of **Section 2** in the original submission.
>
> **Step 3: How does the anchor become a label in SSL**
>
> Based on Step 2, regardless of whether it is G-SSL or D-SSL, the anchor can be regarded as a learning target. Specifically, SSL can be interpreted as follows: In a data augmentation pair, one sample (the anchor) is designated as the target. By constraining the other augmented sample in the feature space to move toward this anchor, consistency in feature representations is achieved. This dynamic adjustment causes samples within the same pair to become tightly clustered, thereby forming an effect similar to a local cluster center.
>
> In traditional classification problems, the common approach is to first project samples into a label space and then constrain them to move toward their corresponding one-hot labels to achieve supervision. In contrast, SSL directly applies constraints in the feature space, which means that the anchor effectively takes on the role of a “label.” In this unsupervised setting, the anchor provides a supervisory signal similar to that of a label. Therefore, it can be argued that labels manifest differently across various spaces—in the feature space, the anchor represents this “implicit label.”
>
> ----
>
> **Response to Weaknesses 3**:
>
> Thank you for pointing this out. We clarify this issue through the following steps:
>
> **Step 1: How we obtain $s$ for a given pair**
>
> $s$ is a vector obtained based on the distribution $q_\phi^e(s \mid x^+, x^{\text{label}})$. In other words, when $\phi$ is given, $s$ is also determined. The learning process of $\phi$ is described in Section 4.1 of the original submission. For a given pair, e.g., $(x_i^+, x_i^{\text{label}})$, we sample once from $q_\phi^e(s \mid x_i^+, x_i^{\text{label}})$ to obtain $s_i$.
>
> **Step 2: The computation of $ba(s_i)$ in Algorithm 1**
>
> Based on Equation (5) in the original submission, we compute $ba(s_i)$. Specifically, this computation is performed with respect to the entire dataset. Given the full training dataset, the pair $(x_i^+, x_i^{\text{label}})$, and $s_i$, the $x_j^{\text{label}}$ involved in Equation (5) are traversed across the entire dataset. Then, according to Definition 4.5, we obtain $ba(s_i)$, which is a vector.
>
> **Step 3: High-level explanation of Algorithm 1 and the identifiability of the spurious variable $s$**
>
> A high-level explanation of **Algorithm 1** is provided in **Appendix F**, while a high-level explanation regarding the identifiability of the spurious variable $s$ is provided in **Appendix G**.

---

### Decision · Program_Chairs · 2025-05-01

**Decision:**

Accept (poster)

**Comment:**

The reviewers generally agree that this paper should be accepted. I concur, albeit on the margins. The reviewers found the theory provided to be solid but raised some questions on both the range of the empirical validation and the generalizability of the proposed method, given notable additional computational needs.

*Strengths.* The theoretical motivations of the work are robust and comprehensive.

*Weaknesses.* The proposed method requires some unverifiable assumptions that may not fully hold in practice. The empirical results give some cover for this, but could be expanded. The proposed method also requires notably more computation resources, which introduces questions about tradeoffs between compute and added performance.

The authors provided clarifications and additional OOD dataset experiments and computational estimates. The set of OOD datasets could be expanded, e.g., Domainbed/WILDS datasets. However, the strengths of the paper outweigh the weaknesses.